# Playing with fire. Understanding how experiencing a fire in an immersive virtual environment affects prevention behavior

Patty C. P. Jansen [ID]1,2¤*, Chris C. P. Snijders2, Martijn C. Willemsen2

1 Department of Marketing Intelligence, Division Interpolis, Achmea Holding B.V., Tilburg, The Netherlands, 2 Department of Human Technology Interaction, Eindhoven University of Technology, Eindhoven, The Netherlands

¤ Current address: Department of Marketing Intelligence, Division Interpolis, Achmea Holding B.V., Tilburg, The Netherlands
* p.c.p.jansen@tue.nl

## Abstract

A potentially effective way to influence people's fire prevention behavior is letting them experience a fire in an immersive virtual environment (IVE). We analyze the effects of experiencing a fire in an IVE (versus an information sheet) on psychological determinants of behavior—knowledge, vulnerability, severity, self-efficacy, and locus of control—based mainly on arguments from Protection Motivation Theory and the Health Belief Model. Crucial in our setup is that we also relate these determinants to actual prevention behavior. Results show that IVE has the hypothesized effects on vulnerability, severity, and self-efficacy, and an unexpected negative effect on knowledge. Only knowledge and vulnerability showed subsequent indirect effects on actual prevention behavior. There remains a direct positive effect of IVE on prevention behavior that cannot be explained by any of the determinants. Our results contradict the implicit assumption that an induced change in these psychological determinants by IVE, necessarily implies a change in behavior. A recommendation for research on the effects of IVE's is, whenever possible, to study the actual target behavior as well.

## Introduction

For the year 2014, the US fire Administration [1] reported 379,500 residential building fires of which 12,075 cases resulted in an injury and 2,765 cases were fatal. The most common cause of these residential building fires were cooking fires [1]. For the Netherlands similar findings apply: a grease fire was the most important cause of fatal fire incidents [2]. The use of prevention measures can help to reduce this number: while a smoke alarm helps to signal a grease fire on time, a fire blanket or a fire extinguisher enables people to extinguish a grease fire before it expands. Insurance companies and society at large would benefit if more people would own and apply preventive measures.

**Data Availability Statement:** All data files are available from the Open Science Framework database. Jansen, P.C.P., Snijders, C.C.P. & Willemsen, M.C. (2020, February 14). Dataset:

effect of IVE on prevention behavior. Retrieved from osf.io/kwq45 DOI 10.17605/OSF.IO/KWQ45.

**Funding:** The first author, Patty Jansen, works three days per week as a marketing researcher at an insurance company (Achmea: https://www.achmea.nl/) and two days per week at Eindhoven University of Technology on her Ph.D. research. Achmea supports her Ph.D. project financially and funded this research. The specific roles of these authors are articulated in the 'author contributions' section. The funders had no role in study design, data collection and analysis, decision to publish, or preparation of the manuscript.

**Competing interests:** Patty Jansen is affiliated with and received financial support from Achmea. This does not alter our adherence to PLOS ONE policies on sharing data and materials.

While mass communication is a typical way to influence people's behavior, mass communication about risks tends to only affect the judgement of risks on a societal level and not on a personal level [3]. In contrast, research concerning natural hazard experiences has shown that personal risk experience can stimulate self-protective and coping behavior [4–6]. Personal experience can influence preventive behaviors, and can even have a larger effect on preventive behaviors than general communication messages because of an increase in susceptibility and worry [7]. A precondition is that the experience is more severe than expected, and that the specific preventive measures are perceived to be effective [7].

In the current study we use an Immersive Virtual Environment (IVE) to have participants experience the large impact of a virtual grease fire and to experience how a simple preventive measure such as a fire blanket can reduce the impact of the fire. Our study has three characteristics that, taken together, are an addition to the current IVE literature. First, while there are studies in other domains that study the effects of experiencing a hazard in an IVE (e.g. flood risk, aircraft evacuation, terror attack), our study is the first to study the effects of experiencing a virtual fire. Second, we consider the effects of an IVE on a full set of "psychological determinants" (knowledge, vulnerability, severity, self-efficacy, and locus of control) and analyze them all together. Whenever we refer to psychological determinants in the remainder of this paper we refer to this set of concepts that are considered drivers for prevention behavior. Third, and most importantly, we measure actual prevention behavior, and test to what extent psychological determinants have an effect on actual behavior. This paper is structured as follows: we first present a brief overview of the use of IVEs to simulate risks. We then present our theoretical framework and hypotheses. Then, we test these hypotheses in a single estimation with a Structural Equation Model. Finally, we discuss the results and their implications.

## Risk simulation in an Immersive Virtual Environment

Simulated experiences can be an effective way to influence people, as people often react to virtual experiences as if they were real [8]. They also allow people to (better) observe the link between cause and effect, which in turn can positively influence peoples' attitudes and behaviors [8]. Next to the persuasive benefits that might arise after the message has been delivered, using virtual reality tools can increase the attractiveness of getting the message across, as people may generally be not interested or motivated enough to search for information themselves.

Studies on the effects of simulated risk experiences in immersive virtual environments (IVE), also known as virtual reality (VR), are relatively recent, although the argument as to why they might work has been around considerably longer. In this paper we also consider virtual 3D environments (the precursors of IVE) as a type of IVE. With IVE "users are perceptually surrounded by and immersed in an artificially-created environment" ([9], p. 57). Often, the user's position and orientation is tracked through a tracking system and the user experiences the virtual environment and his orientation (i.e. if the user's head turns right, visual information of right side of the IVE is perceived) through a head mounted display. Physical movement in the IVE can be performed by making use of advanced tracking systems on the user's body (e.g. special gloves), possibly combined with a motion platform, but also more traditional analog devices (e.g. a joystick) can be used. Currently well-known consumer tools that offer IVE through a head-mounted display are the Oculus Rift, Samsung Gear, and the HTC Vive.

A distinctive characteristic of IVE is its influence on a human's sense of *presence*: "the subjective experience of being in one place or environment, even when one is physically situated in another" ([10], p. 225). This increased sense of presence in IVE can lead to an increase or decrease in emotions, for example an increase of fear when confronted with a virtual spider

[11] or the increase of aggressive feelings in the case of playing a violent videogame [9]. This sense of presence makes, amongst other factors, an IVE also a suitable research tool to study human behavior in a simulated environment, for example to study human behavior while in interaction with fire [12]. Another potential IVE application within the domain of fire, because of the ability to influence emotions and behavior, is to use IVE to influence people's fire prevention behavior.

Several studies have shown that interactive (immersive) virtual environments in which risks are simulated can influence knowledge, emotions, attitudes, and intentions. For example, Zaalberg and Midden [13] have shown that an interactive 3D environment showing a flood simulation can result in increased motivation to search for information, increased motivation to evacuate, and a (small) increase in the willingness to buy flood insurance (compared to non-interactive 2D environments). Furthermore, Chittaro and Zangrando [14] have shown the impact of emotional intensity on awareness of personal fire safety in a fire evacuation game in an IVE. Their results have shown that the highly emotional game (with more visual and audio feedback) produced more anxiety and positively affected participants' attitudes towards the dangers of smoke compared to a more mellow game. Another study of Chittaro and Sioni [15] in which a terror attack was simulated showed that the interactive 3D environment had more impact on risk perceptions than the non-interactive 3D simulation. In the domain of airplane safety, Chittaro [16–18] has shown that a 3D serious game increased the knowledge about safety procedures and feelings of self-efficacy, and made participants feel more "in control" when confronted with emergency landings. This result is suggested to have positive behavioral implications since both self-efficacy as well as safety locus of control have proven to be important predictors for the performance of safety behaviors [16–18]. Also, an IVE can result in more knowledge retention: an IVE in which people experienced a plane crash resulted in more knowledge concerning safety procedures, compared to a safety instruction card, one week after the intervention [19].

While it is promising that previous studies have shown positive effects on knowledge, locus of control, and other psychological determinants, such studies more or less implicitly assume that these effects are indicative of a subsequent improvement in behavior. There are hardly any studies in the safety domain that follow through on the effect of IVE on psychological determinants. In fact, as far as we know, the effects of risk simulation in an IVE on behavior are hardly measured at all, with or without the measurement of psychological determinants. This is the main contribution of our study, applied to the domain of fire prevention.

We now discuss a conceptual model that considers how IVE (when compared to non-IVE delivery of information) affects individuals' psychological determinants with respect to (fire) prevention, and through these, may lead to preventive behavior.

## Theoretical framework and hypotheses

From literature we have identified five psychological determinants that have an important role in influencing individuals' prevention behavior and that are likely to be influenced by experiencing a fire in an IVE: knowledge, severity, vulnerability, self-efficacy, and locus of control. Three of these psychological determinants are grounded in two popular theories that offer a framework for the explanation of prevention behavior: Protection Motivation Theory (PMT) and the Health Belief Model (HMB). The HBM, originally developed by Rosenstock [20] and later discussed and revised by many scholars [21–23], was developed to explain health prevention behavior. PMT was originally developed to explain the effects of fear arousing communications, referred to as "fear appeals" [24], and later extended to a more general theory on behavioral change [25]. Both theories use similar psychological determinants to explain behavior and are commonly used to explain preventive behaviors in the areas of health- and environmental risk. The basic

idea behind both theories is that prevention behavior is driven by the evaluation of the risk (perceived vulnerability and severity), and by an evaluation of the coping response (barriers and benefits of the behavior, self-efficacy). The underlying arguments are equally appropriate for the area of fire prevention behavior, since this contains both a risk element as advised coping responses to deal with the risk. Besides these variables from PMT and HBM, knowledge about the topic and locus of control are also considered variables that may be influenced by IVE and may themselves influence subsequent prevention behavior [17,19,26]. Knowledge about the risk and possible risk-mitigating prevention behaviors, enables people to make the right preventive decisions [27]. While locus of control refers to the extent someone believes a negative event is something within their control, which is related to more safe behaviors [28–31].

The link between these theories and risk simulation in an IVE can easily be established. Research on the effects of IVE's often focuses on exactly the psychological determinants that are considered in PMT and HBM: after the exposure to the IVE someone may feel more vulnerable or at risk of some event *(vulnerability)*, or may assess the consequences of potential events differently than they did before *(severity)* [6,15–17], or may have more confidence in their ability to perform a specific behavior or in their coping ability *(self-efficacy)* [15,16,32–34]. While almost all IVE research focuses on such potential "mental effects", in many cases the ultimate goal is to affect individuals' behavior. However, many researchers have left the relation between psychological determinants and actual behavior untested and sometimes unmentioned, probably in part because it is often difficult, or even impossible, to test the effects of IVE exposure on behavior.

While some studies have demonstrated the relationship between psychological determinants and behaviors [35–37], this relationship is not always straightforward [26] and might depend on the context such as the domain, the behavior under study etc. We argue it is appropriate or at least worthwhile to study psychological determinants and the target behavior(s) simultaneously whenever possible, to test whether a change in determinants really results in a change in behavior.

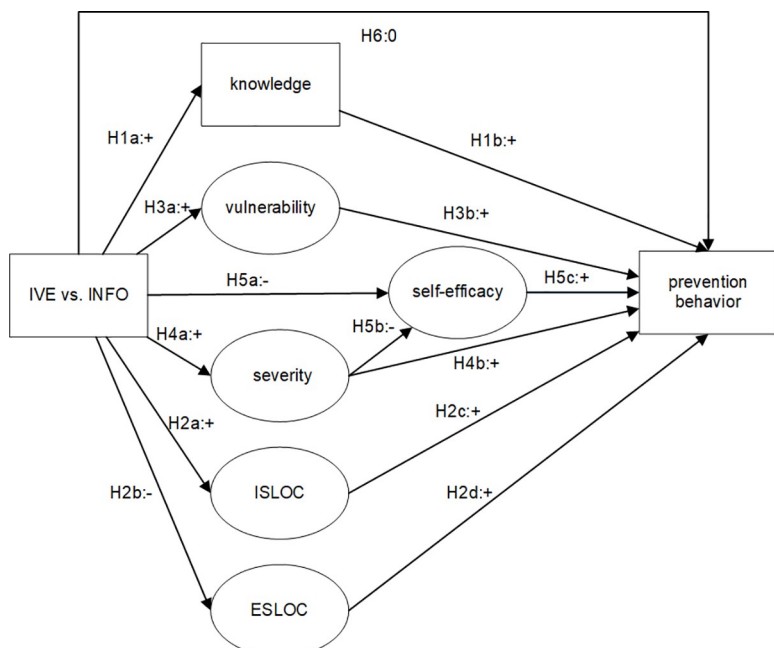

**Fig 1. Hypothesized model of the effects of IVE exposure relative to the INFO condition.** Squares represent observed constructs and oval shapes represent latent constructs.

Based on the arguments as suggested in the literature, and specific to our case of fire prevention, we created a conceptual model that outlines the effects we expect to find (see Fig 1). Although these psychological determinants have been used before and are similar across IVE studies, it still depends on the type of risk under study which determinants are relevant, and in what direction they should be influenced. For example, it is hardly necessary to increase the perceived severity of an emergency landing as people already consider this to be very severe; people should be convinced about their survival possibilities through influencing their knowledge and adoption of the advised safety procedures [16]. On the other hand, in the case of a fire, people are more likely to underestimate the severity, so in this context the aim is to positively influence the perceived severity of a fire [27].

It is important to note another characteristic of our study: we consider all effects of IVE as compared to a control group. Since the traditional way to persuade people to invest in fire prevention measures is by giving them information in text, it is relevant to compare the IVE experience with an information sheet as a control condition (referred to as: INFO condition). We expect that compared to this INFO condition, the IVE has a stronger effect on the psychological determinants, which in turn results in an increased likelihood of performing prevention behaviors in the IVE condition (compared to the INFO condition), as laid out in the hypothesized model in Fig 1. We now discuss the argumentation behind the hypothesized model in more detail for the separate psychological determinants and their effect on prevention behavior.

## Knowledge

Knowledge is considered to be a likely determinant for performing the advised behaviors in a risky situation, especially if one has little time to decide what to do or when in panic [27]. People tend to have limited, or even incorrect, knowledge about fire situations, which makes them insufficiently prepared for a fire [27]. It is evident that to able to select the optimal risk-mitigating actions, one must have knowledge of the risk and be aware of possible actions and their implications (e.g. water cannot be used to extinguish a grease fire but a fire blanket can) [38]. Thus, in order to stimulate people to make the right preventive choices, increasing people's knowledge level of fires and fire prevention might be important. An effective way to sustainably increase knowledge is by an emotional experience [39–41]. Particularly negative experiences enhance the memory of that experience and its details [42]. The literature strongly suggests that serious games can be an effective way for learning [43–45] because games are a good way to attach emotions to problem solving and with that, enhance learning [46]. Several empirical studies have indeed shown positive effects of serious games on knowledge. Kato, Cole, Bradlyn, and Pollock [47] have shown a positive effect of a 3-month serious game on knowledge about cancer treatment, compared to a control group who received a commercial game video game (i.e. Indiana Jones). One-time interventions have also proven to work: Wong et al. [48] have shown that a serious game is better in transferring factual knowledge compared to traditional textbooks, both directly after the intervention as one week later. A study of Chittaro [19] has also shown positive effects of a serious aircraft evacuation game in an IVE on knowledge retention. One week after the intervention the players of the game had a significantly higher score on the knowledge items than the group who had read a safety card, even though immediately after the intervention there was no difference between the groups. Furthermore, Chittaro [32] showed that an interactive mobile VR application teaching how to don a life preserver had positive effects on donning a life preserver in real life, comparing a traditional safety card.

Consequently, we expect that a serious game in an IVE, one that triggers emotions and increased attention by a virtual fire, can positively influence people's knowledge level about

fire prevention. We also expect that an increase in knowledge positively influences prevention behavior.

H1a: The knowledge level of participants in the IVE condition is higher than the knowledge level of participants in the INFO condition.

H1b: Knowledge positively influences prevention behavior.

## Safety Locus of Control (SLOC)

Locus of control is an important psychological determinant that can influence our attitudes and behaviors [49] and for that reason it is often studied in situations where the goal is to change people's behavior. Locus of control is our perception of where control lies, and most researchers distinguish between an internal and an external orientation [31,50,51]. Someone with an internal orientation is more likely to consider outcomes as a consequence of their own behavior, as opposed to someone with an external orientation, who often sees outcomes as something beyond their own control. In the literature there is no consensus about the dimensionality of the locus of control construct [50,52]. Although the original scale of Rotter [53] presents a one-dimensional construct (where internal and external are two ends of a continuum), later studies reveal two (with internal and external being two separate constructs) or even multiple dimensions. Most studies that we came across find empirically that locus of control is a two-dimensional construct, rather than the more intuitive one-dimensional construct [31,50,51]. The underlying argument is that people for example can attribute their health both to internal beliefs (for example related to their own smoking behavior) as well as to external beliefs (for example to chance events). An internal locus of control is associated with safer attitudes and behaviors: the more someone feels that he or she is responsible for how matters are progressing, the more likely that someone will take action. Inversely, an external locus of control is associated with a lack of caution and prevention. Studies have indeed shown that, for instance, people with an internal locus of control compared to people with an external locus of control are more likely to wear seat belts [28] and have less accidents at work [29,30]. Furthermore, studies showed that internal control is negatively related to fatal car accidents while external locus of control is positively related to fatal car accidents [31]. Huang and Ford [36] have shown in addition that locus of control is not a fixed human characteristic but can be influenced by training, as they have demonstrated with respect to safe driving behaviors. Murray, Fox, and Pettifer [54] have shown that a higher sense of realism in a virtual environment can likewise lead to an increased perception of locus of control. Also Ahn, Bailenson, and Park [26] managed to influence environmental locus of control with an IVE in which deforestation was at stake. Furthermore, in the domain of air safety Chittaro [17] showed that a videogame that focused on the brace position during a plane crash resulted in a significant increase in internal orientation and a significant decrease in external orientation, which together indicated that participants felt more in control over the outcomes of an emergency landing than before playing the game.

As far as we know, locus of control has not yet been analyzed in the fire safety domain, but it makes sense that it is relevant in that domain as well. We follow the line of reasoning that safety locus of control (SLOC) is a two-dimensional construct, existing of an internal orientation (ISLOC) and an external orientation (ESLOC). We expect ISLOC and ESLOC to be affected more for IVE than for the INFO condition and expect changes in ISLOC and ESLOC to positively influence behavior.

H2a: Perceived ISLOC is higher for participants in the IVE condition than for participants in the INFO condition.

H2b: Perceived ESLOC is lower for participants in the IVE condition than for participants in the INFO condition.

H2c: Perceived ISLOC positively influences prevention behavior.

H2d: Perceived ESLOC positively influences prevention behavior.

## Vulnerability

An important determinant to take preventive measures according to the HBM and PMT, is the likelihood that an event will occur, referred to as the perceived vulnerability (or: suscepti-bility) [20,24]. A common barrier to take preventive measures is that people tend to think that bad things will not happen to them, a phenomenon known as 'unrealistic optimism' [7]. A way to increase people's perceived vulnerability is through personal experience [7]. A study of Zaal-berg et al. [6] has shown that people who have experienced a flood before, perceive themselves as more vulnerable to a future flood. This higher perceived vulnerability, together with a higher perceived effectiveness for adaptive actions, made that flood-victims had more intentions to take adaptive actions (e.g. tie up or remove curtains to prevent them from getting wet) than non-victims. However, no effect was found with respect to preventive measures (e.g. sandbags in front of the house). This might be explained by the fact that victims perceived the adaptive actions to be more effective than non-victims, while the effectiveness for preventive actions was found more effective by non-victims. That is, increased vulnerability leads to more pre-ventive or adaptive measures, provided that the measures are considered effective enough.

Since we want to prevent people to become victims in the first place, we could increase the perceived vulnerability through the *simulation* of a risk. This has also been done by Schwebel et al. [55] who studied the effects of the virtual simulation of crossing a street while texting, and Chittaro [15–18] who measured the effects of the simulation of an aircraft emergency and a terror attack. While the IVE's concerning street crossing and a terror attack showed signifi-cant positive effects on perceived vulnerability [15,55], the IVE's that simulated an aircraft emergency [16–18] showed no significant effects. While the perceived probability of an aircraft evacuation might not increase through a virtual experience given that most people fly on air-planes relatively rarely, affecting people's perceived vulnerability to a fire is more likely, as most people cook several times a week, or use lighters, candles etc. much more regularly. We expect that the virtual experience of a fire increases people's perceived vulnerability to a fire, which in turn will result in more preventive behaviors.

H3a: The perceived vulnerability of participants in the IVE condition is higher than in the INFO condition.

H3b: Perceived vulnerability positively influences prevention behavior.

## Severity

Another important determinant for taking preventive measures according to the HBM and PMT, is the perceived severity of the event and its consequences [20,24]. The direction of this relationship depends on the expected severity beforehand, since some experiences appear to be milder than expected [7]. With respect to flood experiences, research has shown that the experience of a flood did increase the perceived severity of a future flood, compared to people who did not have this experience [6]. Chittaro [17,18] managed to increase perceived severity in an aircraft evacuation game using rich and vivid feedback to induce fear, although the effect was small. In the domain of fire safety, increasing perceived severity is even more relevant (an

airplane crash is already considered quite severe) since with fires in buildings people seem to downplay the severity and as a consequence do not respond quickly enough, which decreases their chances of survival [27]. Chittaro and Zangrando [14] argue that people's fire evacuation behavior can be influenced by increasing anxiety and risk perception. Their study showed that a game about the dangerous effects of smoke had more impact on anxiety and attitude when emotional intensity was high (as it might be in an IVE). We therefore expect that a fire in an IVE can similarly increase perceived severity, which itself is considered a relevant psychological determinant for prevention behavior.

H4a: The perceived severity of a fire by participants in the IVE condition is higher than in the INFO condition.

H4b: Perceived severity positively influences prevention behavior.

## Self-efficacy

Another important determinant to take preventive measures that is related to the person (instead of related to the risk) according to the extended versions of the PMT and the HBM is self-efficacy [23,25], a construct that is related to locus of control. "Perceived self-efficacy is the belief in one's competence to tackle difficult or novel tasks and to cope with adversity in specific demanding situations" ([56], p. 81). In this sense, self-efficacy is different from locus of control: it does not consider the extent to which a person feels that the situation depends on his or her own behavior, but instead how well a person perceives he or she would perform in the part that does depend on his or her behavior. According to Bandura [57] optimistic efficacy beliefs lead to better performance outcomes, those beliefs can be acquired by gaining experience, and this experience can be gained through an IVE [8]. For example, in the healthcare domain, serious games for children, compared to a control group, have shown positive effects on self-efficacy in taking care of a chronical condition [58,59]. Also in the safety domain similar effects are found: a serious aircraft evacuation game has shown positive effects on self-efficacy in safely evacuating an aircraft [16,18] and a mobile VR application concerning life preserver donning increased people's self-efficacy [32]. Sometimes, instead of pursuing an increase in people's self-efficacy level, it is desirable to decrease self-efficacy. For instance, in driving studies people tend to overestimate their driving ability [60] and in this case a higher perceived self-efficacy is related to more unsafe driving behaviors and accidents [61–63]. This is comparable to the fire prevention domain where people generally tend to overestimate their ability to evacuate, because people downplay the severity of a fire [27,64]. In general, people start moving too late and too slowly in the case of a fire, and often even move through smoke while this should be avoided as this slows down their speed and is dangerous for their health. The suggested negative relationship between perceived severity and self-efficacy has been empirically shown in the health domain [65,66]. Since we expect that people underestimate the consequences of a grease fire, and overestimate their own ability to act properly, we expect that the IVE can decrease perceived self-efficacy with respect to a grease fire, both directly as well as through an increase in perceived severity and that a lower self-efficacy will motivate people to take more fire prevention measures.

H5a: The perceived self-efficacy of participants in the IVE condition is lower than in the INFO condition.

H5b: Perceived severity negatively influences self-efficacy.

H5c: Perceived self-efficacy negatively influences prevention behavior.

## Benefits and barriers of behavior

Other determinants that drive prevention behavior according to the HBM and PMT are the benefits of the behavior (more specifically: its effectivity) and the barriers of the behavior (e.g. financial costs, effort). Because in our experiment a fire blanket is promoted as an effective measure to extinguish a grease fire in both conditions, we do not expect 'effectivity' to be influenced more in the IVE condition and therefore do not include effectivity in our study. Because the IVE and INFO condition did not stress the costs and effort involved concerning a fire blanket, we also do not expect costs and effort to play an important role in our study, and therefore do not include these determinants in our study.

## Prevention behavior

The ultimate goal of the simulation of a fire in an IVE is to increase people's fire prevention behavior. Most studies have only considered the effect of IVE on psychological determinants and sometimes on intentions but did not measure subsequent behavior. This implies we do not have much empirical support for an effect of IVE on prevention behavior. However, if our earlier hypotheses about the effects of IVE on psychological determinants are valid and changes in psychological determinants lead to changes in behavior, then the logical consequence is that IVE influences behavior.

H6: Because of the changes in psychological determinants caused by the IVE condition, we expect that participants in the IVE condition display more prevention behaviors than participants in the INFO condition.

## Retention of the psychological determinants

Most IVE-related effect studies only consider effects immediately following the intervention. However, in some cases longer term effects on knowledge have been studied and have led to positive results for IVE. In Chittaro [19] we find that while there no was no immediate effect of a serious aircraft evacuation game on knowledge, the players of the game had a significantly higher score on knowledge than the control group one week later. Furthermore, Wong et al. [48] have shown that a serious game was better in transferring factual knowledge than a traditional textbook, and this effect was still demonstrable one week later. The sometimes implicit and largely intuitive argument for this effect is that because people have an increased sense of presence in an IVE (compared to people who read an information sheet), their experiences make a more lasting impression, which improves retention. Along these lines, we expect a higher retention rate (or slower decrease of retention) of the effects of the IVE on the psychological determinants.

# Method

## Design

We used a 2 factor (IVE versus INFO) between-subjects design, with 3 time points at which we asked the participant to fill out a questionnaire. The first questionnaire was right before the intervention, the second questionnaire right after the intervention and the third questionnaire four weeks after the intervention. Our main intervention has two levels: the IVE fire game (IVE) and a fire prevention information sheet (INFO) as a control.

 

## Participants

Participants were recruited from the Dutch commercial "CG Selections" consumer panel, a panel consisting of 80,000 consumers. We determined that we would need 2 x 119 participants to have 90% power of showing an expected difference in prevention behavior of about twenty percentage points (assuming 20% for the INFO group vs 40% for the IVE group; using an alpha-level of 5%).The prevention behavior that we targeted was whether a participant would invest part of his or her show-up fee in a fire blanket and whether a participant would take home flyers related to fire safety. We recruited participants based on the following criteria: between 18 and 70 years old, not living in a student home or living with their parents (due to prevention responsibility), who did not own a fire blanket and who did not suffer from motion sickness (to prevent them from getting nauseous during the IVE). CG Selections approached potential participants based on the age and location information in their database with the opportunity to participate in a study about "property damage" for an incentive of €25. We did not mention the topic "fire prevention" to prevent any bias (e.g. attract people that have a special interest in fire; people becoming extra aware of their own fire prevention). To determine whether people fitted the required criteria to engage in the study they first had to fill out a survey with questions concerning their living situation, motion sickness, and whether they owned a fire blanket. Because we did not want to prime people with the fire blanket, we confronted people with a list of possible home appliances and asked which products they owned.

After enrolling, participants were randomly assigned to the IVE group or the INFO group. Each day was either an IVE day or an INFO day, and participants were assigned to a condition based on availability. In total 297 participants started the experiment, of which 49 were omitted from the analysis because in the first questionnaire (right before the intervention) they reported to own a fire blanket, three were excluded because they failed to fill out the second questionnaire and three others were excluded because they did not complete the IVE due to nausea. In total 242 participants remained in the dataset for analysis.

## Procedures

The experiment was approved by the ethical committee of the Department Human Technology Interaction of the Technical University of Eindhoven. The experiment ran in the lab for nine days. In total nine persons assisted the first author in running the experiment. When entering the lab, first the participant was asked to read and sign a written informed consent. Second, we asked participants to fill out the first online questionnaire. Then, dependent on the assigned condition, the participant was asked to play the IVE game or read the INFO sheet. Participants in INFO condition received fire prevention information on a single page A4 (S1 File) with basically the same content as the experiences in the IVE contained (S2 File). The intervention lasted about 8 minutes for the IVE group and 2 minutes for the INFO group. To evaluate the IVE game, participants were asked afterwards whether they became nauseous or dizzy, how realistic they considered the fire experience, and how severe they considered this virtual experience. Then, participants in both groups were asked to fill out a second online questionnaire. Then, we offered the participant a choice of receiving their promised €25 or receiving €12.50 and a certified fire blanket with a value of €20. The money would be transferred after they filled out the third questionnaire (four weeks later), the fire blanket could be taken home immediately. In addition, to measure the interest for fire prevention information, there were two types of fire prevention related flyers on the table, which participants could take home. To avoid social desirability bias the availability of these flyers was not specifically mentioned by the experimenter. Four weeks after the participant was exposed to the intervention, the participant received the third online questionnaire.

 

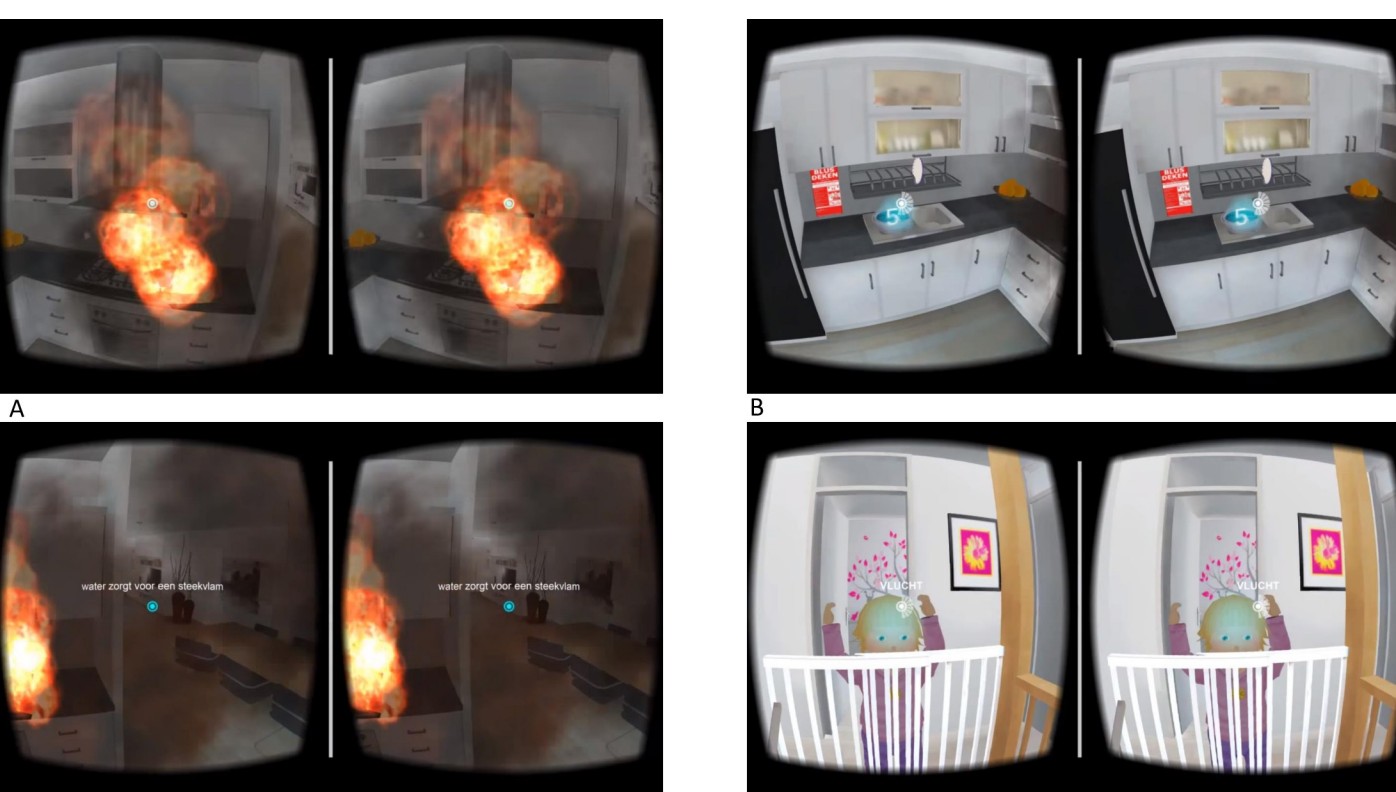

**Fig 2. Screenshots of the IVE fire game.** (A) View of the grease fire. (B) View of the bucket of water and the fire blanket that can be selected to extinguish the fire. (C) View of the flash fire. (D) View of the toddler that is waiting to be rescued.

### The IVE condition and participant's actions

In the IVE condition, participants were told that they were going to do a fire drill in a virtual environment. The participant received a game controller, a headphone, and a head-mounted-display. It was explicitly stated that if the participant became nauseous or otherwise uncomfortable, he or she could stop at any time. First, the simulation showed a practice scene so that the participant could get acquainted with the virtual environment. During the practice scene it was explained to the participant how to move in the environment (walk through the house, go up- or downstairs) and how to select an action (use water or fire blanket, open doors, pick-up toddler). The participant can move (walk, turn) in the IVE by using the joystick of the controller and choose different actions by focusing on the objects with his eyes. If the participant focusses on an object (e.g. the fire blanket), a blue lightbulb appears that is counting down from 5 seconds to 1 second, after which the action is performed (e.g. use fire blanket to extinguish fire). When the participant indicated to be ready for it, the actual game was started. See Fig 2 for screenshots of the IVE fire game.

At the start of the game the participant (in the IVE) is situated on the couch in the living room, watching television. Suddenly the smoke detector goes off, and smoke is entering the living room. The participant can 1.) go towards the source of the fire (the kitchen) and extinguish the fire with the fire blanket. The fire will be extinguished and the game ends. Or 2.) go towards the source of the fire (the kitchen) and extinguish the fire with water. In this case a flash fire occurs and the participant has to escape the house to survive (using the fire blanket at this point will not help any more). A final option was 3.) to escape from the home immediately

through the front door. The participant could however also not perform action 1, 2 or 3, but stay in the house. Thirty seconds after the smoke detector goes off, the fire gets bigger as the flames now hit the kitchen hood and cabinets. At this point, if the participant is still in the house, he will hear a toddler crying upstairs, and he could choose to go upstairs and rescue the child. After the participant had either extinguished the fire, escaped the home or 105 seconds had passed, the game ended. Dependent on the decision the participant made, textual feedback and information about fire prevention was presented. For instance, if the participant had not been able to extinguish the fire and was still in the house, a text would explain that in case of a fire in all likelihood three minutes is the maximum one has to escape, and that proper use of the fire blanket can prevent a lot of harm. See S2 File for the possible scenarios and the feedback given after each scenario.

After playing the game for the first time, participants were asked to play the game once again. If they had extinguished the fire with the fire blanket the first time, they were now suggested to play the game without using the fire blanket (in which case the optimal strategy would to try to escape). If they did not extinguish the fire with the fire blanket the first time, they were asked to try to extinguish the fire with the fire blanket. This way, all participants experienced both escaping and extinguishing a fire.

## Hardware

We used the Oculus Rift DK2 Head-Mounted Display (HMD) with a resolution of 960 x 1080 per eye and a 100 degrees diagonal field of view. Audio was played through Sennheiser HD 265 linear headphones. For the study, the software was implemented on one desktop machine and one laptop. Given our setup, the specifications of the hardware needed to be substantial, given current day hardware. The desktop machine was a Mac Pro 3.7GHz, with 16GB of RAM, a 256GB Flash Drive and 2x AMD Fire Pro D300 graphics card with 2GB each. The laptop was a MacBook Pro 2.4Ghz, with 8 Gb of RAM, 1 TB HDD, and a GeForce video card with 512MB.

## Measurements

For this study we used measurements at three time points. The first measurement was a questionnaire right before the intervention that included background questions and current prevention behaviors that are beyond the score of this paper. Background questions included family composition, type of home, ownership of home, year of construction, frequency of cooking, frequency of playing computer games and experience with home fire(s). The second measurement took place right after the intervention and was set up for the measurement of the psychological determinants and the target prevention behaviors. The hypothesized model (Fig 1) is based on differences between the IVE and the INFO group at the time of the second measurement. The third measurement, four weeks after the intervention, was set up to be able to analyze the development of the psychological determinants over time. In this measurement, the determinants *knowledge*, *vulnerability* and *severity* were included, and we also included the same prevention behaviors as in the first measurement. We excluded self-efficacy, ISLOC and ESLOC from the third measurement because of lengthiness, which could negatively affect responses to the (at-home) questionnaire and reliability of the answers. The prevention behaviors that were measured in the first and third measurement are not analyzed in this paper.

**Knowledge.** Knowledge concerning fire prevention was measured with nine self-constructed items that were related to the information provided in the IVE fire game and the INFO sheet. Example items are: *What happens when you throw water on a grease fire*? and *On average, how many minutes does a person have to safely leave the home in case of a fire*? See S3

File for an overview of all items. Correct answers were coded 1 and incorrect answers were coded 0. The overall knowledge score was the percentage of correctly answered items.

**Internal and external safety locus of control.** We designed the locus of control scale specific for fire prevention, since a domain-specific locus of control scale is a better predictor for domain specific behavior than the general locus of control scale [67]. In line with Chittaro [17] locus of control was measured with 12 items, which were adopted from Hunter [51], and were adapted to fit the topic of fire. There were 6 items internally oriented (ISLOC) and 6 items externally oriented (ESLOC). What complicates matters somewhat is that locus of control can refer to two separate matters in the event of a fire: whether someone feels that he or she can influence the probability of a fire, and whether one can influence the consequences of a fire. We formulated items for both matters. An example of an internally oriented item that was focused on preventing a fire was: *If you are careful, you can prevent fire in your home*. An example of an externally oriented item that was focused on reducing the consequences of a fire was: *If there is a fire in your home, there is usually nothing you can do*. Ratings were given on a five-point Likert scale (1 = fully disagree; 5 = fully agree).

**Vulnerability.** Vulnerability was measured with 3 items. In line with Chittaro [16,17] we adopted the items from de Hoog, Stroebe, and de Wit [68], and transferred the items to the subject of a grease fire. Ratings were given on a five-point Likert scale (1 = very low; 5 = very high). An example item was: *How high do you perceive the risk of a grease fire in your home to be?*

**Severity.** Severity was measured with 3 items. In line with [16,17] we again adopted the items from de Hoog et al. [68] and transferred the items to the subject of a grease fire. Ratings were given on a 5-point Likert scale (e.g. 1 = not severe at all; 5 = very severe). An example item was: *How severe do you perceive the consequences of a grease fire to be?* We changed one item of original scale, namely concerning the seriousness of the grease fire, since in Dutch this would also be translated as "severe". We changed the item to "panic" since this also reflects the severity of the grease fire.

**Self-efficacy.** Self-efficacy was measured with 10 items. The items were adopted from the General Self-Efficacy Scale (GSE) [69] and from the scale that Chittaro [16] used to measure self-efficacy with respect to aircraft evacuation (which was also based on the GSE). All 10 items were transferred to the domain of fire prevention behavior, following [70] who argues self-efficacy should be tailored to the specific domain of functioning to have the best explanatory and predictive value. Ratings were given on a five-point Likert scale (1 = fully disagree; 5 = fully agree). An example items was: *I am confident that I can extinguish a grease fire.*

**Specific prevention behavior.** In both conditions of the experiment a fire blanket is the advised behavior to extinguish a grease fire. The primary measurement for the effect of IVE on prevention behavior will be the participants' purchase of a fire blanket, and the secondary measurement is the participants' interest in fire prevention information (which is a more 'soft' measurement of behavior). We measured the purchase of a fire blanket by offering the participant a choice between the promised incentive of €25, or €12.50 and a fire blanket for participation. Concerning fire prevention information, there were two types of prevention related flyers on the table that participants could take home. We registered (unobtrusively) whether a participant chose the fire blanket (0 = no fire blanket; 1 = fire blanket) and whether the participant took one (or more) flyers (= no flyers; 1 = flyers).

**Additional measurements.** For the experiment we only recruited people that reported not to own a fire blanket. To verify, we asked again if people had a fire blanket in the first questionnaire. If so, they were excluded from the analysis. Furthermore, we included several variables that we used for evaluation purposes and robustness checks: we verbally asked the IVE group right after the IVE to what extent they became nauseous or dizzy, how realistic they considered the fire experience, and how severe they considered the virtual experience.

## Results

First we briefly describe our preparation of the data and then show some descriptive statistics. To answer our research question with respect to the impact of IVE on the psychological determinants and the actual prevention behavior, we use Structural Equation Modelling (SEM). Finally, we analyze the development of the psychological determinants between the second and the third measurement.

### Data preparation

There were no missing values in the first and the second questionnaire, except as a consequence of conditional items. However, 10.6% of the participants (26 out of 242) did not fill out the third (at home) questionnaire. We checked the data for multivariate outliers with the BACON algorithm using Stata 14 [71] and did not find any. We checked the data for normality with a skewness and kurtosis test [72] and the Shapiro-Wilk test [73] and found that not all variables were normally distributed so we used a robust estimator for non-normally distributed data. For SEM it is important that items or latent constructs do not correlate too much with each other [74] and there were no bivariate correlations larger than .85. Detecting multicollinearity among multiple variables was done by considering the variance inflation factors (VIF) after a logistic regression with the choice of a fire blanket as the target variable and all scale constructs included. This assumption was not violated: all variables have a VIF $<10$ and together have a mean VIF of 1.20 [73].

### Descriptive statistics

Of the remaining 242 participants in our analysis, 124 were assigned to the IVE condition, and 118 to the INFO condition. Of these 242, 146 females (60.3%) and 96 males (39.7%). Mean age equaled 42.5 (SD = 10.11). A chi-square test and a two-sample Wilcoxon rank-sum test (age was non-normally distributed) showed that there was no significant difference between the IVE and INFO condition in terms of gender and age respectively ($\chi^2$ (1, $N$ = 242) = .703, $p$ = .402; $z$ = .164, $p$ = .870).

After the IVE game, 21.8% indicated that they had felt nauseous or dizzy during the experience, 32.3% indicated that they felt a little nauseous or dizzy, and 46.0% did not feel nauseous or dizzy. Participants rated the realism and severity of the IVE experience on a five point Likert scale (where a five stands for very realistic / severe). The mean scores were 3.82 (SD = .97) for realism and 3.43 (SD = 1.11) for severity. In the IVE condition, 48.4% of the participants chose the fire blanket and €12.50 and in the INFO condition this was 39.8% ($\chi^2$ (1, $N$ = 242) = 1.795, $p$ = .180). In the IVE condition, 23.1% of the participants took the flyers home and in the INFO condition this was 12.8% ($\chi^2$ (1, $N$ = 242) = 4.179, $p$ = .041). So there is some evidence for a relation between the IVE manipulation and prevention behavior, but the effect is smaller than we assumed (a twenty percent point difference) when calculating our sample size. The specific actions that people took in the IVE fire game can be found in S1 Table.

### Statistical analyses

We tested the model in Fig 1 through SEM with a robust estimator for non-normally distributed data. We used SEM because we wanted to determine the relations between the observed and latent constructs in a single estimation. An alternative possibility to test the hypothesized model is to analyze the model with SEM in three steps, namely first test the total effect, then the effects of IVE on the psychological determinants followed by the effects of the determinants on prevention behavior, in line with a standard mediation analysis. Or, we could

separately show the effect of VR on the psychological determinants, and only then the (separate) effects of the psychological determinants on behavior. We found that these separate analyses show substantially the same results: all hypotheses of interest remain statistically significant and of similar size. For parsimonious reasons, we present the SEM model that tests all the relationships in a single estimation.

Some further beneficial features of SEM are that it is possible to use both observed and unobserved (i.e. latent) variables, and that the measurement error of observed and unobserved variables are taken into account simultaneously. Different compared to most experimental IVE studies is that with SEM the model as a whole is being tested instead of a set of individual hypotheses, including the role the psychological determinants might play in the effect of IVE on prevention behavior. The SEM model was analyzed in Mplus version 8 [75]. The estimation procedure that we used was the default weighted least squares with means and variance adjusted (WLSMV, which implies using probit as the underlying analytical model) since this is considered the best estimator for categorical data [74,76]. Whether the model fits the data can be determined through the model fit statistics and the individual parameter estimates. As advised in Hair, Black, Babin, and Anderson [77] we report the following model fit statistics: model chi-square, root mean square error of approximation (RMSEA) and its 90% confidence interval [78], the comparative fit index (CFI) [79] and the Tucker Lewis Index (TLI) [80]. The higher the model chi-square the worse the fit, and the associated level of significance must be non-significant [77]. For the RMSEA Hu and Bentler [79] advise a value smaller than .06 for a good fit, with the upper bound of its 90% CI falling below 0.10 [81]. For the CFI and the TLI, Hu and Bentler [79] advise a value larger than .95.

## Confirmatory factor analysis

The latent constructs were analyzed by CFA in a single estimation, so that correlations between (items of) latent constructs could be taken into account [81,82]. When all items of the latent factors (vulnerability, severity, self-efficacy, ISLOC and ESOC) were used, model fit statistics were poor: $\chi^2$ (340) = 1080.666, $p$ = < .001, CFI = .875, TLI = .861, RMSEA = .095, 90% CI [.089 - .101]. In order to establish construct validity we inspected the significance levels of the items, the direction of the estimates, the item estimates and R-squares (83). Convergent validity and a more precise measurement is established by removing items based on the AVE, which ideally has to be larger than .5 [77]. Indeed, inspecting the CFA we find that many items have low R-squares and that the AVE of the constructs was low (below .5 for some). We increased the AVE for self-efficacy from .560 to .682 by removing five items, for ISLOC we increased the AVE from .364 to .413 by removing three items, and for ESLOC we increased the AVE from .343 to .549 by removing four items. Since ESLOC only had two items left, and a minimum of three indicators per latent variable is needed for model identification [82] the correlation between ISLOC and ESLOC was very high (-.792, suggestion discriminant validity is compromised as the correlation is higher than the square root of the AVE of ISLOC itself), and the items seem to fit well on one scale, we decided to use a one factor SLOC instead. When performing CFA with all latent constructs and the latent factor SLOC, the same results apply concerning item deletion, as with CFA with all latent constructs and the latent factors ISLOC and ESLOC. Internally orientated items (I) contribute positively to SLOC, while externally orientated items (E) contribute negatively to SLOC. Although in literature two factors are more common, LOC was originally developed as a one-dimensional construct and is still used as such by various studies [26,53,83]. Table 1 shows the final AVE and Cronbach's alpha values of the latent factors, and the factor loadings per item. The model fit then improved to $\chi^2$ (98) = 478.448, $p$ = < .001, CFI = .914, TLI = .895, RMSEA = .127, 90% CI [.115 - .138]. The

**Table 1. AVE, Cronbach's alpha and standardized factor loadings (stdYX) for the latent factors, after removing poorly fitting items.** Items without a factor loading where excluded from analysis.

| Latent factor[a] | Item[b] | Factor loading[c] (stdYX) |
|---|---|---|
| Vulnerability | How high do you think the risk of a grease fire in your home is? | .570 |
| α = .745 | How high do you perceive the chance that a grease fire will pass | .660 |
| AVE = .584 | over to the exhaust hood and the kitchen cabinets? | |
| | How high do you perceive the chance that you should escape your home because of a grease fire? | 1.002 |
| Severity | How dangerous do you think a grease fire is? | .829 |
| α = .752 | How severe do you perceive the consequences of a grease fire? | .851 |
| AVE = .705 | How much panic do you think there will be in case of a grease fire? | .840 |
| Self-efficacy | I am confident that I can extinguish a grease fire. | .834 |
| α = .834 | I am confident I will remain calm in case of a grease fire. | .848 |
| AVE = .682 | I am confident I will remain calm in case of a grease fire, even if it will pass over to the exhaust hood and the kitchen cabinets. | |
| | When a grease fire exists I am afraid I will panic. [d] | .782 |
| | I know what to do in case of a grease fire. | .772 |
| | I am capable of acting correctly in case of a grease fire. | .889 |
| | I am convinced of my capability to put my family/ myself into safety in case of a fire. | |
| | I am convinced of my capability to quickly leave my home in case of a fire. | |
| | I am convinced of my capability to quickly leave my home in case of a fire, even if escape routes are blocked. | |
| | I am convinced of my capability to quickly leave my home in case of a fire, even if there is a lot of smoke. | |
| Safety Locus of | If you are careful, you can prevent a fire in your home yourself. (I)[c] | |
| Control (SLOC) | If a fire breaks out in your home, there is usually nothing that you | |
| α = .710 | can do. (E)[e] | |
| AVE = .528 | A home fire is usually caused by a short-circuit/ overheating of | |
| | electrical appliances. (E) | |
| | Whether people can escape in time in case of a fire, is a matter of luck or bad luck, not of preparation. (E) | |
| | People can ensure that a small fire does not expand. (I) | .653 |
| | Most home fires are caused by chance events that cannot be influenced. (E) | |
| | Preparing yourself for a fire, enlarges your survival possibilities in case of a fire. (I) | .661 |
| | Whether you succeed in extinguishing a grease fire, is a matter of luck or bad luck, not a matter of preparation. (E) | -.777 |
| | People should be rewarded by their insurance company if they take preventive measures to prevent or control a fire. (I) | |
| | By taking preventive measures you can make sure that you can extinguish a fire on time. (I) | .733 |
| | Home fires are usually caused by the people themselves. (I) | |
| | It has no use to prepare yourself for a fire in your home. (E) | -.796 |

[a] Latent factor originates from the second measurement.

[b] Items are translated from Dutch.

[c] Factor loadings are only present for items that were not removed from the scales.

[d] This item was reversed coded.

[e] The (I) refers to an internally orientated item. The (E) refers to an externally oriented item.

factor loadings and the Cronbach's alpha values of the original scales can be found in S4 File, Table 1. In addition, we performed CFA on the individual factors and similar results apply compared to testing the measurement models all together, and the same items should be removed to improve AVE. See Tables A-E in S4 File for the model fit statistics and Tables F and G in S4 File for the R-squared estimates and AVE's.

## Structural Equation Models

We now test the model as depicted in Fig 1. Table 2 shows the results of fitting six different models, each slight variations of the base model from Fig 1. The models differ in terms of the changes that were made to improve model fit, but do not change in terms of the significance and size of the effects for our underlying hypotheses. That is, the general conclusions with respect to our hypotheses do not depend on, which model is used, underscoring the robustness of our findings.

Model fit statistics of the initial model (Fig 1, not in Table 2) were poor: $\chi^2$ (155) = 807.382, p = < .001, CFI = .840, TLI = .804, RMSEA = .132, 90% CI [.123 - .141]. Since IVE did not significantly affect SLOC and SLOC did not relate to prevention behavior, we removed SLOC from further analyses. We then tested the model as proposed in Fig 1 again without SLOC *(model 1)*. Model fit statistics improved but were still not adequate, as can be seen in Table 2. We further improved the model fit, as suggested by the mod-indices, by adding correlations between latent variables that we did not hypothesize a-priori *(model 2) (severity ↔ vulnerability; self-efficacy ↔ vulnerability)*. There relations are not illogical since for example vulnerability and severity together are assumed to represent a larger construct referred to as "risk perception" [84] or "threat appraisal" in Protection Motivation Theory [24,25]. Also we added correlations between items *(model 3) (self-efficacy ↔ sev15; self 6 ↔ self5; self6 ↔ self1)*, as these relations were suggested by a mod-indices analysis. We only incorporated this step to show how model fit could be improved, but will not elaborate on this step, as adding correlations at the item level is not very common. A logical next step, after model 2, is to inspect the significant relations and remove the non-significant ones *(model 4)*. Some relations that were hypothesized a priori appeared to be non-significant and were removed, of which many represented the effect of psychological determinants on behavior *(vulnerability → fire blanket; severity → fire blanket; self-efficacy → fire blanket; knowledge → flyers; severity → flyers; self-efficacy → flyers; IVE → flyers)*. Model fit indeed improved compared to model 2. For a visual representation of model 4 see Fig 3. Next, we removed the non-significant relations from model 3 and found that the model fit further increases but the relations themselves do not change *(model 5)*. We then elaborated on model 4 and removed all constructs that have no direct or indirect effect on the target variables (fire blanket or flyers) *(Model 6)*. Although this procedure results in a better model fit, the relationships between the observed and latent constructs remain the same, which underscores the robustness of our findings. Model fit statistics are then very good. For a visual representation of model 6, see Fig 4.

## Robustness checks

We tested the hypothesized relationships also on variations of the original sample to check the robustness of the outcomes. We tested all the relationships on the sample minus the participants who stated they became nauseous during the IVE experience (n = 215), on the sample with only the IVE participants who stated to find the experience (very) realistic (n = 199), and on the sample with only the IVE participants who stated to find the experience (very) severe (n = 187). With the exception of some small deviations in model fit statistics and standardized

**Table 2. Model fit statistics, standardized regression weights [stdYX] and standard errors [S.E.] for model 1–6.**

| Model fit statistics | | | | | | | |
|---|---|---|---|---|---|---|---|
| Goodness of fit | Target values | Model 1 | Model 2 | Model 3 | Model 4 | Model 5 | Model 6 |
| $\chi^2$ | | 590.867 | 408.292 | 150.458 | 390.103 | 148.994 | 15.753 |
| df | | 78 | 76 | 73 | 81 | 78 | 12 |
| $p$ | >.05 | < .001 | < .001 | < .001 | < .001 | < .001 | .203 |
| RMSEA | < .06 | .165 | .134 | .066 | .126 | .061 | .036 |
| 90% CI | < .10 | .153-.177 | .122-.147 | .051-.081 | .113-.138 | .046-.076 | .000-.079 |
| CFI | >.95 | .854 | .905 | .978 | .912 | .980 | .994 |
| TLI | >.95 | .803 | .869 | .968 | .886 | .973 | .990 |
| Relationships | | | | | | | |
| Independent variabele | Dependent variabele | StdYX [S.E.] | StdYX [S.E.] | StdYX [S.E.] | StdYX [S.E.] | StdYX [S.E.] | StdYX [S.E.] |
| IVE | knowledge | -.333[.060] | -.333 ***[.060] | -.333***[.060] | -.333***[.060] | -0.333***[.060] | -.333***[.060] |
| IVE | vulnerability | .229**[.069] | .231**[.067] | .231 **[.068] | .240***[.067] | .240***[.067] | .239**[.069] |
| IVE | severity | .169*[.069] | .169*[.069] | .158*[.071] | .169*[.069] | .158*[.071] | |
| IVE | self-efficacy | -.231***[.064] | -.230***[.064] | -.276***[.066] | -.230***[.064] | -.276***[.066] | |
| severity | self-efficacy | -.387***[.054] | -.388***[.054] | -.189 **[.068] | -.388***[.054] | -.189**[.068] | |
| knowledge | fire-blanket | .225**[.086] | .225**[.086] | .225**[.086] | .225**[.086] | .225**[.086] | .225**[.086] |
| vulnerability | fire-blanket | .124[.123] | .091[.102] | .078[.103] | | | |
| severity | fire-blanket | .136[.118] | .102[.101] | .096[.095] | | | |
| self-efficacy | fire-blanket | .049[.118] | .074[.107] | .033[.102] | | | |
| knowledge | flyers | .079[.659] | .225[.086] | .010[.084] | | | |
| vulnerability | flyers | .321*[.138] | .246*[.110] | .245*[.111] | .251*[.103] | .252*[.103] | 0.264**[.102] |
| severity | flyers | -.024[.121] | -.057[.099] | -.052[.094] | | | |
| self-efficacy | flyers | -.143[.144] | -.032[.128] | -.030[.128] | | | |
| IVE | fire-blanket | .154[.092] | .167[.088] | .160[.088] | .183* [.084] | .183*[.084] | .183*[.084] |
| IVE | flyers | .228[.220] | .143[.100] | .142[.101] | | | |
| Independent variable | Independent variable | | | | | | |
| fire blanket | flyers | .488 ***[.107] | .490 **[.106] | .490**[.101] | .505***[.108] | .505***[.108] | .507***[.108] |
| severity | vulnerability | | .292***[.059] | .287***[.060] | .289***[.059] | .285***[.061] | |
| self-efficacy | vulnerability | | -.324***[.058] | -.395***.[55] | -.325***[.058] | -.395***[.055] | |

Model 1: as proposed in Fig 1 but without SLOC

Model 2: as model 1 and with correlations between the latent variables

Model 3: as model 2 and with three correlations on an item level

Model 4: as model 2 but without the non-significant relationships

Model 5: as model 3 but without the non-significant relationships

Model 6: as model 4 but without all relationships that did not have an [in]direct effect on the target variables

* = $p < .05$

** = $p < .01$

*** = $p < .001$

regression weights, results showed that all estimated relationships remained stable across the different samples.

## Hypotheses

Tests of the hypothesized relationships can be found in Table 2 and we will address them here in more detail based on Model 4 (Fig 3). Contrary to what was hypothesized (H1a), the IVE fire game results in a lower knowledge level than the INFO condition (-.333, $p < .001$). Thus,

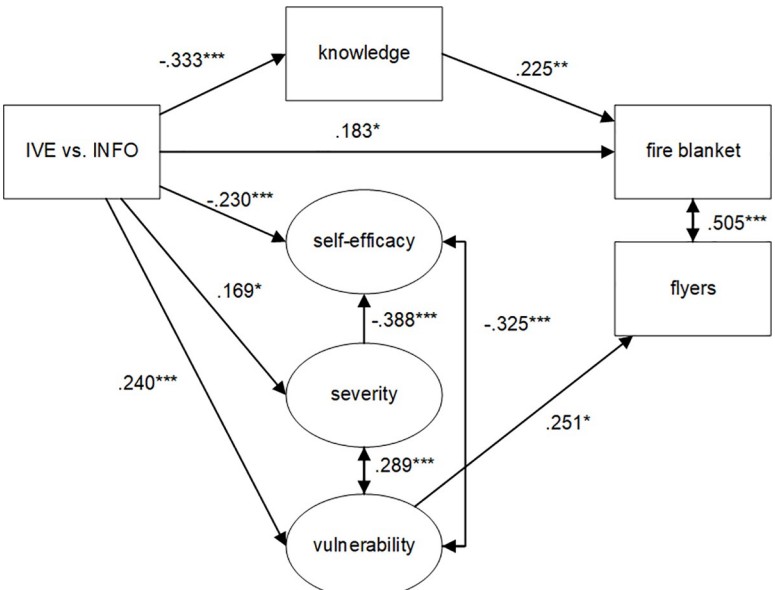

**Fig 3. Structural Equation Model of model 4 with standardized regression weights [StdYX].** Squares represent observed constructs and oval shapes represent latent constructs. Significance levels: * = *p* < .05; ** = *p* < .01; *** = *p* < .001.

participants who were provided the text sheet knew more about fire prevention than participants who experienced the virtual fire and were provided with this same information in the IVE. As expected, an increase in knowledge implies a higher probability to choose the fire blanket (H1b) (.225, *p* = .009). H2a-H2d concerned ISLOC and ESLOC. Based on our results we integrated ISLOC and ESLOC into a single concept: SLOC, and as a consequence we could only test H2a and H2c. Further analyses showed that participants who experienced the virtual fire did not have a higher internal safety locus of control compared to participants who were provided with the text sheet. Also, a higher level of internal safety locus of control did not correlate with prevention behavior. Therefore, H2a and H2c are rejected. H3a is supported: the vulnerability of participants in the IVE condition is higher than the vulnerability of participants in the INFO condition (.240, *p* < .001). H3b can be partly confirmed, as an increase in vulnerability only influenced taking home flyers (.251, *p* = .015), but did not affected participants' choice for the fire blanket. H4a is supported: the severity of a fire by participants in the IVE condition is higher than the severity by participants in the INFO condition (.169, *p* =

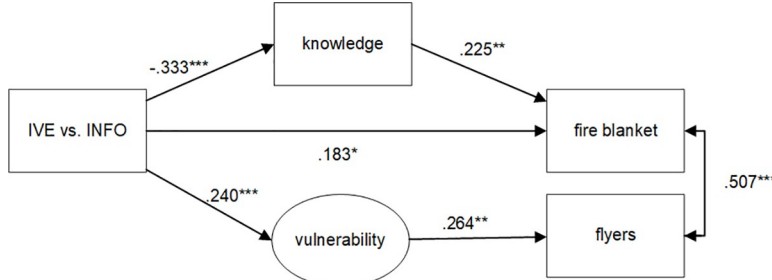

**Fig 4. Structural Equation Model of model 6 with standardized regression weights [StdYX].** Squares represent observed constructs and oval shapes represent latent constructs. Significance levels: * = *p* < .05; ** = *p* < .01; *** = *p* < .001.

.014). However, an increase in severity did not influence prevention behavior (H4b). As hypothesized in H5a, the self-efficacy of participants in the IVE condition is lower than the self-efficacy of participants in the INFO condition (-.230, $p < .001$). Also, a higher severity negatively influenced self-efficacy (H5b) (-.388, $p < .001$). However, a lower self-efficacy did not affect prevention behavior (H5c).

Strictly speaking, H6 is rejected, but some careful consideration is necessary here. To test H6 we now distinguish the total, direct, and indirect effects based on model 6 from Table 2 (Fig 4), using a percentile bootstrap estimation approach with 1000 samples [85]. All coefficients are presented as standardized regression weights (*stdXY*). The total effect of IVE on fire blanket consists of a significant direct effect of IVE on the fire blanket (b = .197, SE = .081, 95% CI [.065, .726], $p = .016$) and a significant indirect effect via knowledge (b = -.080, SE = .030, 95% CI [-.279, -.044], $p = .007$). The total effect, however, is non-significant (b = .116, SE = .078, 95% CI [-.082, .550], $p = .136$). Hence, the non-significant total effect of IVE on fire blanket appears to be a combination of two significant opposite effects (positive and negative). Additionally, the effect of IVE on flyers is the consequence of (only) an indirect effect via vulnerability. There is a positive and significant effect of IVE on vulnerability (see Model 6, Table 2) and a positive and significant effect of vulnerability on flyers (see Model 6, Table 2), but the total effect is only significant at the p = .052 level (b = .063, SE = .032, 95% CI [-.002, 254], $p = .052$). Moreover, H6 implies that any effect of IVE on prevention behavior will be mediated by the psychological determinants. This part of the hypothesis is clearly rejected for the fire blanket, as there remains a direct effect of IVE on choosing the fire blanket.

## Retention after four weeks

We now consider the development of the scores on the psychological determinants, to test whether the IVE has a positive effect on retention. Results are analyzed with a multilevel regression analysis with robust errors, correcting for the non-normality of the residuals.

On average, the knowledge level had decreased after 4 weeks, as there was a significant difference on knowledge between the first measurement (M = .86, SD = .13) and the second measurement (M = .78, SD = .13; b = —.038, $p < .001$). This decrease in knowledge is much stronger for the INFO group (from M = .91, SD = .09 to M = .79, SD = .13) than for the IVE group (from M = .82, SD = .14 to M = .78, SD = .13), as reflected in a statistically significant interaction (b = -.096, $p < .001$). For a graphical illustration of these differences, see Fig 5.

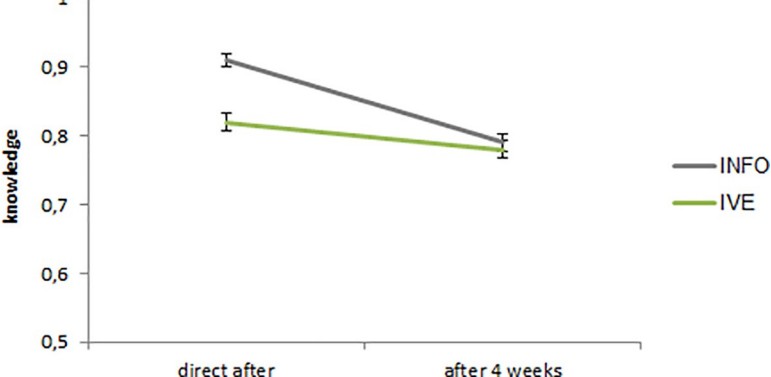

**Fig 5. Knowledge level of the IVE and the INFO condition in the first and second measurement.** The error bars represent the standard error (SE).

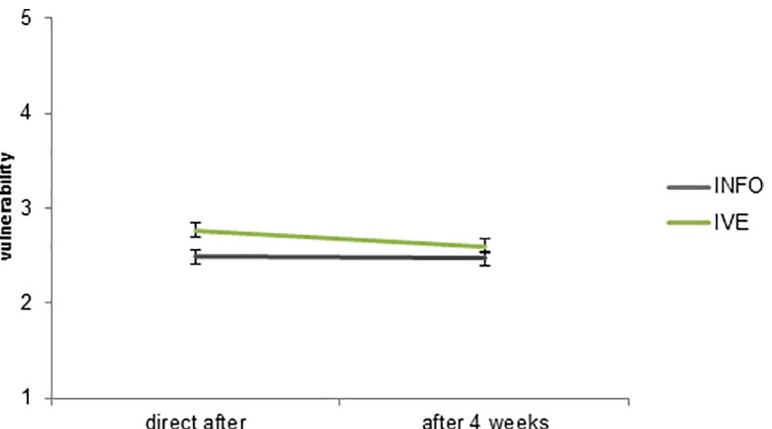

**Fig 6. Vulnerability level of the IVE and INFO condition in the first and second measurement.** The error bars represent the standard error (SE).

On average, the vulnerability level was lower after 4 weeks, as there was a significant difference with respect to vulnerability between the first measurement (M = 2.64, SD = .83) and the second measurement (M = 2.54, SD = .82; b = .327, $p$ = .002). There was no significant interaction effect between IVE and vulnerability (b = .1636, $p$ = .085) (see Fig 6).

On average, the severity level was slightly lower after four weeks, but there was no significant difference on severity between the first measurement (M = 3.73, SD = .62) and the second measurement (M = 3.63, SD = .61; b = -.062, $p$ = .218). There was no significant interaction effect between IVE and severity (b = -.063, $p$ = .379) (see Fig 7).

Whereas previous results in the literature have suggested increased retention in IVE, our results support this only to a limited extent. First, vulnerability decreases somewhat over the span of four weeks, but we do not find substantial differences between the IVE and the INFO condition. Severity does not significantly decreases over time. There is a difference over time in terms of knowledge retention. After four weeks, the higher knowledge that the participants had in the INFO condition has decreased to the level of the IVE condition, which itself hardly decreased over the four weeks. In this sense, there is some evidence for an IVE leading to better retention than INFO in terms of knowledge. However, in our case the IVE condition had a lower knowledge level to begin with.

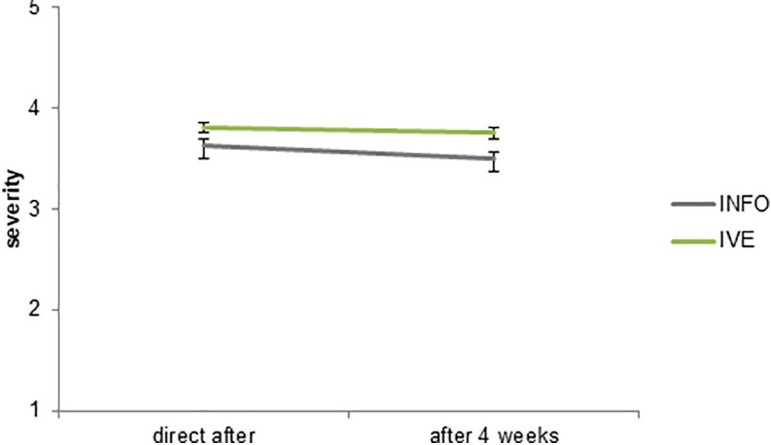

**Fig 7. Severity level of the IVE and INFO condition in the first and the second measurement.** The error bars represent the standard error (SE).

## Discussion

We considered the effects of an IVE on relevant and frequently used determinants for prevention behavior—knowledge, vulnerability, severity, self-efficacy, and locus of control—, and the extent to which these effects in turn affect subsequent prevention behavior. The main reason for our approach was that most IVE studies only consider the effect of the IVE on these psychological determinants [13,14,16–19,34,86], but not its effects on actual prevention behavior, nor the mediating effects of the psychological determinants.

The IVE had effects on almost all of the psychological determinants. As expected, the vulnerability and severity levels were higher in the IVE condition compared to the INFO condition: people in the IVE group felt more vulnerable to fire and thought a grease fire was more severe. After four weeks these feelings were approximately equally present in both groups. Also, as expected and desired, the level of self-efficacy was lower in the IVE condition compared to the INFO condition. That is, people felt less confident that they would act properly in case of a grease fire. This is partly a direct effect, and partly an indirect effect through the increase of the perceived severity of a grease fire. An unexpected result was that people in the INFO condition scored higher on knowledge. This could possibly be explained by the fact that the information in the IVE was presented as text in the head-mounted display, which may not have been so easy to read and perhaps also not compatible enough with the rest of the virtual experience. Also, in the INFO condition people were really focused on the information, as their only task was to read the presented information, while in the IVE condition people were more focused on the fire and on how to act and interact. One could imagine that the effect of the IVE could be improved by changing the way in which information is put forward. For instance, improved software might make it easier to produce a static text that is easier to read than the one in our IVE was, or by getting the information across through sound. Also, instead of showing the information afterwards, it might work more effectively to integrate all knowledge components in the IVE game (as in: [19,32]). Even though this was done for some knowledge components (e.g. the text"water causes a flash fire" was shown after the participant used water to extinguish the grease fire), other knowledge components were only presented at the end. Although the INFO group scored higher on knowledge directly after the intervention, their knowledge level significantly decreased after 4 weeks and decreased more rapidly than in the IVE group. This suggests that knowledge presented in a traditional text format is not retained over a longer time span as well as information in an IVE, as in line with results of Chittaro and Buttussi [19]. Safety locus of control (SLOC) was not influenced by the IVE at all, even though effects of interventions (e.g. IVE, training) on locus of control have been found in other domains [17,36]. A possible reason for this might be that locus of control is hard to influence in the fire prevention domain, possibly because the score of SLOC was already quite high (the mean score of the SLOC items equaled 4.3 were all items were measured on a five point scale).

The IVE did affect the measured prevention behaviors, however it affected different behaviors in different ways. When only comparing the subsequent behavior and the original conditions, IVE did not have the desired positive effect on investing in a fire blanket, compared to a control condition in which the same information was delivered on paper. However, closer inspection of this result shows that this effect was a combination of a direct positive effect of IVE and an indirect negative effect via a decrease in knowledge. One could potentially argue that the negative effect of IVE on knowledge could have had consequences for the other psychological determinants, however we did not came across these relationships in literature, nor did the mod indices in our SEM models suggest this. IVE had a (small) effect on vulnerability and vulnerability had a (small) effect on taking home flyers. Taken together this lead to a

(smaller) total effect of IVE on taking home flyers that was no longer significant in our sample. It is somewhat unexpected and noteworthy that the IVE affects these prevention behaviors in different ways. This highlights the importance of a better understanding of the causal mechanisms that might lead individuals to change their behavior. In this case we found that a higher knowledge level results in investing in a fire blanket, but not to more information seeking through taking home flyers. More knowledge about the risk, the available actions and its consequences results in the fire blanket as the perceived optimal choice. The explanation for no increase in flyers might be that because one already has an increased knowledge level, there is no interest in further information. On the other hand, we found that a higher vulnerability did lead to more information seeking. Perhaps the higher vulnerability is a sign that participants realized that they knew less about the risks that they thought they did, and therefore wanted to get more information, which is understandable, although we arrive at this only in hindsight. It remains to be seen whether this connection between vulnerability and information seeking holds across a broader set of domains. Nevertheless, it seems sensible for future research to categorize the prevention behaviors in terms of the kind of psychological determinant it is affected by, instead of assuming that all prevention behaviors are equally affected by the psychological determinants.

The IVE affected knowledge, vulnerability, severity, and self-efficacy but the latter two did not relate significantly to prevention behavior, despite the fact that previous literature has indicated these to be potentially important determinants of behavior. The effects on the prevention behaviors are only affected by two of the psychological determinants: knowledge and vulnerability, and both are determinants for *different* prevention behaviors. Moreover, the effect of IVE on investing in a fire blanket is not fully mediated by the decrease in knowledge, but is for the most part a direct effect that cannot be explained away by any of the psychological determinants that we considered. This raises the question which other factors there might be that could explain the direct relationship between IVE and prevention behavior that we did not take into account. One possibility would be to consider the other factors of the HBM and the PMT that we did not include here, namely whether the perceived benefits and barriers related to the prevention behavior in question might nevertheless have played a role [20,24]. One of these variables is for example the perceived effectiveness of the prevention behavior, an important determinant, together with vulnerability, for taking adaptive actions to minimize the consequences of a flood [6]. Another question that arises because of the results of this study is how to interpret the effect of IVE on psychological determinants in other studies. Does an increase in for instance vulnerability or severity through IVE necessarily imply an actual behavioral change? Especially in fields in which the specific psychological determinants under study have not been validated with real behavioral outcomes (e.g. aircraft evacuation, flood experience, fire) our analyses suggests that one must be careful with the interpretation and generalization of such results.

## Limitations

As the focus of this study was on fire and fire prevention, it remains to be seen whether results are generalizable to other areas of risk. For example, SLOC was not influenced by the IVE in our study, and did not affect the measured prevention behaviors, while studies in other domains did show these results.

Furthermore, we only measured two kinds of behavior: investing in a fire blanket and taking home flyers. As we have seen, direct and indirect effects may differ depending on the measured behavior and we do not know whether and how results would differ had we included additional prevention behaviors. For example, maybe an increase in the perceived severity

would have influenced the probability of purchasing a fire extinguisher or would have led to paying more attention to the fire escape plan in the building.

In addition, one could argue about the appropriateness of the INFO condition as the control condition, as it differs on multiple aspects from the IVE condition (e.g. occupation time, no graphics, no gaming element). An alternative would have been to complement the current setup with a more comparable condition such as the 2D version of the same fire game, although this would probably lead to smaller sized effects. Our power analysis showed that we needed about 240 participants, so adding an additional condition would already dramatically increase the number of participants. Moreover, if the effect size would indeed be smaller given that the two conditions are more similar, this would imply a still larger necessary sample size.

We could also have measured the psychological determinants before the experimental manipulations, so that they could be used as control variables in our analyses, which might have increased precision and could have provided a baseline measurement against which to compare post-intervention measurements. The trade-off, however, is that we would be priming the participants in the direction of effects on these determinants by the intervention (possibly making it 'easier' to find differences). We therefore chose not to measure the determinants beforehand.

A further useful addition could have been to include the perceived effectiveness of the fire blanket as a measurement, since this is perceived to be an important determinant for performing prevention behaviors [7]. However, we did not included this question since we did not want to prime participants too much in the direction of buying the fire blanket.

## Conclusion

Although the IVE influenced most of the psychological determinants, not all of these psychological determinants subsequently influenced the target prevention behaviors. The effect of the IVE on investing in the fire blanket was partly mediated by knowledge (and partly a direct effect), and the effect on taking home the prevention flyers was fully mediated by an increase in vulnerability. Given that we have included all psychological determinants that we found in the literature, it is surprising that the larger part of the effect of IVE does not seem to be correlated with these determinants. This suggests three issues that are noteworthy for IVE research in general. First, merely that only establishing effects of IVE on psychological determinants such as vulnerability and locus of control (as is usually the case in IVE studies) does not necessarily imply effects on prevention behaviors as not all psychological determinants lead to subsequent prevention behavior. Second, different prevention behaviors can be influenced by different psychological determinants: there is a real need to consider in more detail which determinants trigger which kinds of prevention behaviors. Finally, our study shows that not all effects of IVE on prevention behavior can be "explained away" by the psychological determinants that find their origins in the HBM and PMT and are typically measured in IVE research. Taken together, these two findings should be cause for some concern with regard to the often used setup in IVE research, where only the effect of IVE on psychological determinants is measured.

## Supporting information

**S1 File. Text for INFO group (translated from Dutch).**
(DOCX)

**S2 File. Text in IVE (translated from Dutch).**
(DOCX)

**S3 File. Knowledge scale (translated from Dutch).**
(DOCX)

**S4 File. Additional results of Confirmatory Factor Analysis.**
(DOCX)

**S1 Table. Actions people took in the IVE fire game, during the first and second game play.**
(DOCX)

## Acknowledgments

We thank the colleagues from Interpolis who provided their expertise, support and assistance during the research. We thank Purple, for the development of the IVE and help with the technical details during the experiment. Also we would like to thank the two anonymous reviewers whose comments and questions helped to improve and clarify this manuscript.

## Author Contributions

**Conceptualization:** Patty C. P. Jansen, Chris C. P. Snijders, Martijn C. Willemsen.

**Data curation:** Patty C. P. Jansen, Chris C. P. Snijders, Martijn C. Willemsen.

**Formal analysis:** Patty C. P. Jansen, Chris C. P. Snijders, Martijn C. Willemsen.

**Funding acquisition:** Patty C. P. Jansen.

**Investigation:** Patty C. P. Jansen.

**Methodology:** Patty C. P. Jansen, Chris C. P. Snijders, Martijn C. Willemsen.

**Project administration:** Patty C. P. Jansen.

**Resources:** Patty C. P. Jansen.

**Supervision:** Chris C. P. Snijders, Martijn C. Willemsen.

**Visualization:** Patty C. P. Jansen.

**Writing – original draft:** Patty C. P. Jansen.

**Writing – review & editing:** Patty C. P. Jansen, Chris C. P. Snijders, Martijn C. Willemsen.

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
