## [Decision Letter · Decision Letter 0]

19 Aug 2019

PONE-D-19-17627

Playing with fire. Understanding how experiencing a fire in an immersive virtual environment affects prevention behavior.

PLOS ONE

Dear Ms. Jansen,

Thank you for submitting your manuscript to PLOS ONE. After careful consideration, we feel that it has merit but does not fully meet PLOS ONE’s publication criteria as it currently stands. Therefore, we invite you to submit a revised version of the manuscript that addresses the points raised during the review process.

We would appreciate receiving your revised manuscript by Oct 03 2019 11:59PM. To enhance the reproducibility of your results, we recommend that if applicable you deposit your laboratory protocols in protocols.io, where a protocol can be assigned its own identifier (DOI) such that it can be cited independently in the future. For instructions see: http://journals.plos.org/plosone/s/submission-guidelines#loc-laboratory-protocols

We look forward to receiving your revised manuscript.

Kind regards,

Geilson Lima Santana, M.D., Ph.D.

Academic Editor

PLOS ONE

Journal Requirements:

In the interest of full disclosure, we wish to draw your attention to the following: the first author, Patty Jansen, works three days per week as a marketing researcher at an insurance company (Achmea) and two days per week at Eindhoven University of Technology on her Ph.D. research. Her employer contributes partly to her Ph.D. research by financially supporting her for one day per week.  The employer also financially supported this research. We declare that the research that we present was in no way influenced by the insurance company. The funder had no role in study design, data collection and analysis, decision to publish, or preparation of the manuscript.

https://www.achmea.nl/

We note that you received funding from a commercial source: Achmea

4. "We note from your Cover Letter and Financial Disclosure that the first author of the manuscript (FCPJ) is affiliated to an insurance company. Please update the affiliations within your Cover Page and within Editorial Manager to reflect this affiliation.

Reviewers' comments:

Reviewer's Responses to Questions

**Comments to the Author**

1. Is the manuscript technically sound, and do the data support the conclusions?

Reviewer #1: Partly

Reviewer #2: Partly

2. Has the statistical analysis been performed appropriately and rigorously? 

Reviewer #1: Yes

Reviewer #2: No

3. Have the authors made all data underlying the findings in their manuscript fully available?

Reviewer #1: No

Reviewer #2: Yes

4. Is the manuscript presented in an intelligible fashion and written in standard English?

Reviewer #1: Yes

Reviewer #2: Yes

5. Review Comments to the Author

Reviewer #1: I’ve read the paper with interest. It’s about a timely topic (comparing VR and non-VR approaches for teaching safety behaviors) and aims at exploring an additional variable with respect to other studies.

The way the paper is written does not yet meet the requirements of an high-quality presentation, so I provide a number of comments in the following about necessary improvements, that I hope we will be helpful in preparing a thoroughly revised version.

First, the paper needs to tone down its initial claims of extreme originality (three of the four claims are exaggerated).

“Our study has four characteristics that, taken together, make it stand out from most previous IVE studies.” COMMENT: Actually this claim is not accurate and must be toned down for three of the four characteristics.

“First, our study focuses on the fire domain, while most risk related IVE-effect studies

involve other risk domains (e.g. health, traffic safety, environmental risk, aircraft evacuation).” COMMENT: Several IVEs concerning fires have actually been built, see this scholar search as a starting point:

https://scholar.google.com/scholar?hl=en&as_sdt=0%2C5&q=%22virtual+reality%22+Fire&btnG=

“Second, we consider the effects of IVE on a full set of “psychological determinants

(knowledge, vulnerability, severity, self-efficacy, and locus of control). “ COMMENT: Also effects of virtual risk experiences on such variables have been previously studied (and the paper should contrast its research with the other findings on these psychological variables), for example:

Chittaro L., Designing Serious Games for Safety Education: "Learn to Brace" vs. Traditional Pictorials for Aircraft Passengers, IEEE Transactions on Visualization and Computer Graphics, 22, 2016, 1527-1539.

Chittaro L., Sioni R., Serious Games for Emergency Preparedness: Evaluation of an Interactive vs. a Non-Interactive Simulation of a Terror Attack, Computers in Human Behavior, 50, 2015, 508–519.

Both these papers also refer to the same theoretical basis of the submitted manuscript (Protection Motivation Theory, PMT)

“Third, we compare the effects of IVE with a control condition which allows establishing the effect of IVE over and above a more standard way of getting the same information across.” COMMENT: See above

“Fourth, and most importantly, we measure actual prevention behavior, and test to what extent psychological determinants have an effect on actual behavior.” COMMENT: This is the actual key point to stress. It’s the characteristic on which this research stands out.

However, the authors have to be careful about terminology concerning behavior. The two behaviors measured in the study were 1) if a participant would invest part of his or her show-up fee in a fire blanket and 2) whether a participant would take home flyers related to fire safety. There’s a terminological issue all over the paper (that can however be easily fixed): the paper uses the term “prevention behaviors” to refer to the two behaviors, but what they actually show is a possible interest in prevention (for example, taking home a flyer is no guarantee that the participant will keep and read the flyer and take the preventive actions described in the flyer).

The paper (correctly) stresses the importance of VR presence creating the expectation that the construct will be measured in the study, but then it is not. This is a bit confusing.

According to PMT, a simple hypothesis that Perceived Vulnerability and Perceived Severity positively influence behavior, as the paper does (H3b and H4b) is not straightforwardly supported. The theory indeed clearly states that if vulnerability and severity are perceived, but the recommended behavior is not presented in a way that convinces the message recipient about efficacy (both recommendation efficacy and self-efficacy), then the influence will be negative instead of positive: the recipient will engage in emotion control behaviors instead of the recommended risk control (prevention) behavior.

Hypothesis “H5c: Perceived self-efficacy negatively influences prevention behavior” needs more theoretical motivation. As it is, it apparently contradicts the theory (PMT) on which the paper aims at being substantially grounded.

The description of the IVE condition is lacking of enough details. For example, the paper says “The participant can then choose between different actions: 1.) go towards the source of the fire (the kitchen) and extinguish the fire with the fire blanket. The fire will be extinguished and the game ends. Or 2.) go towards…”

A first question is how such actions are selected? Second, the granularity of the actions is not clear. For example, what happens if one selects action 1? All those listed things happen as the user passively watches? Or does the user have to perform them? In this latter case, there are multiple actions. And what does the user do in the physical world to perform the multiple actions?

More generally, since one of the goals of the IVE in the paper is to increase users’ knowledge, it is important to fully describe everything users could do and what was the feedback from the environment, also highlighting the pedagogical aspects of the IVE design.

Attaching videos to the submission could provide some additional help.

In the hardware section, the paper says the software was implemented on one

desktop machine and one laptop. But what happened during the study? Some people used the laptop and some other the desktop?

The measurement section is not always clear too. The first measurement is said to “include current prevention behaviors that are beyond the score (sic) of this paper.” What does this mean? Were people asked about their fire prevention behaviors?

Then the paper says “The prevention behaviors that were measured in the first and third measurement are not analyzed in this paper”. One can try to guess that this was done to assess if there was a behavior change (at least a self-reported ones). If this is the case, it’s strange that the paper omits to analyze it, since the main difference from other research in the literature concerns specifically participant’s behavior.

To measure the SLOC construct, the paper creates a questionnaire in which some items are supposed to measure internal SLOC while the others external SLOC. However, the section does not report a factor analysis to confirm that the set of created items can actually be divided in the two subscales.

One of the items (“How much panic do you think there will be in case of a grease fire?”) that were used to measure severity is different from the usual items that measure that construct, and the choice should be explained.

One of the items (“People should be rewarded by their insurance company if they take

preventive measures to prevent or control fire”) that were used to measure SLOC does not actually concern locus of control orientation.

Unlike the first two questionnaires, the third questionnaire was administered on-line. The paper does not provide details about actions taken to prevent issues of on-line surveys, for example: were the answering times measured and taken care of ?

The “descriptive statistics” says that the effect of the IVE is smaller than expected, but provides no effect sizes in the statistics.

In Table 1, the translations from Dutch to English is not always good, in one case “Fire in a home usually exists by the people themselves” the reader cannot be sure of what that is supposed to mean.

The percentage of participants who felt nauseous during the VR experience is surprisingly high and needs discussion. It’s an indication that some VR design guidelines were likely violated by the implemented IVE. This can limit significantly the effectiveness of the IVE for the goal intended in the paper.

The result that participants using VR learned less than those using printed information needs much more discussion. From the details and screenshots provided, a simple and likely explanation can be traced back to the design and implementation of the IVE, which seems too primitive compared with the state of the art in the VR literature. The paper should discuss the differences in interactivity, realism, and pedagogical methods between the IVE used and the more sophisticated IVEs available in the literature.

Moreover, since the IVE can influence psychological determinants both indirectly (through level of knowledge) and directly, the limitations in the IVE can have had consequences that reach furthest than knowledge only, and weaken the conclusions of the paper. This should be discussed in a critical way.

Care must also be taken in the conclusions section which currently seems to assume that the results hold for IVEs in general, while the limitations in the studied IVE make it difficult to support generalization to more complex IVEs used in learning and training.

Reviewer #2: In the present manuscript, the authors provide an interesting approach towards fire prevention by using an immersive virtual environment procedure to teach people how to react in the case of a grease fire. The inclusion of actual prevention behavior, together with self-report measures and the use of an experimental setting with randomization and control condition gives this article a promising potential. However, there are some writing and data-analytic issues that, in my opinion, need to be addressed before a recommendation for publication. In my review, I will focus mostly in consistency, writing form and methods which is my area of expertise.

The statistical analyses (SEM) done in this article do not take full advantage of the experimental setting created by the authors, given that the results are not controlled by baseline levels (as in a group x time interaction model), which is a problem the authors should address in detail. It is a common practice to measure the constructs in a pre-post fashion, however the authors did not do it to avoid priming. However, the trade-off is that there is no baseline control anymore, turning the analyses correlational, which should be addressed by the authors in more depth.

On the other hand, I find the data analysis and results section rather confusing. This is, in my opinion, mostly because it is very data dense and contains a lot of modelling, which always creates a challenge in reporting. One consequence of this, is the omission of important information to evaluate the quality of the data analyses. This is especially relevant for the CFA and SEM parts, where they show some inconsistencies that I’m listing below.

pp 6 (and others). When talking about psychological determinants, please be more specific. Psychological determinants of prevention behavior?

pp 7: "The underlying arguments are equally appropriate for the area of fire prevention behavior" It is important to give arguments when presenting such a statement, why?

Pp 7. "Studies usually use" is not the best argument for selection of variables: I strongly suggest the authors to describe a sound logical and empirical basis for their proposal. This is done later; however, it can be briefly explicated here.

Pp.7 "We feel it is appropriate or at least worthwhile to study psychological determinants and the target behaviors simultaneously" Why? It is, indeed, important, but this should be elaborated.

pp 8. "Can in fact be studied empirically relatively easily" How?

pp11. Please check repeated citation (29,29).

pp11. Please provide references for the SLOC bi-dimentionality, if they were provided before, I suggest moving the definition of the construct together with the citations above.

pp 22. Why are only knowledge, vulnerability and severity included in the third measurement? This exclusion should be supported.

Measures: Please report basic psychometric properties of the instruments here (cronbach's alpha for internal consistency, for example, and factor structure of original instruments).

pp. 27. Please specify the software version used for SEM analyses.

Data analyses and report:

Please describe which sampling wave was used for the analyses (I guess it’s sample 2, however I cannot find it in the manuscript).

Measurement models: I find it problematic to test all measurement models together, given that this would further obscure local sources of bad fit. Is it just one instrument the one creating the bad fit? Are all measurement models wrong? I would suggest test this separately. On the other hand, the procedure of co-variation between items within an instrument should be done in this step.

Structural Equation Models: Modifications were made to the models, which is a step towards a good fit. It is important to see if the modifications suggested by the modification indexes are reasonable under theoretical grounds. This is done for the co-variation between variables; however, it should also be made for the covariation between items. It is usually accepted to co-vary items within a scale if they share method or another known source of shared variance that should not be captured by the latent factor (please see previous comment, however, regarding the location of this procedure).

It is not clear why do the authors base their report on model 4, if the model with a good fit is model 6. It is important to report the good fitting model. Even though models showed no substantial changes in their point estimates, the non-significant results are part of a bad-fitting model and thus are not directly interpretable (i.e. SLOC). This should be at least stated as a limitation or tested with a different regression technique.

I find it important to explicitly say, on the SEM models, how were the dichotomous outcomes treated, and interpret them accordingly. Where they estimated using a probit or logistic regression? (In Mplus this is different for different estimation methods)? Please clarify this on the manuscript.

Please also describe which method did the authors use to compute the mediation analyses (multiplicative?) and provide the estimates in the tables.

Pp 36. Please specify which bootstrap method was used. Bias corrected, percentile? They have shown different trade-offs regarding type 1 errors.

Pp 36. A direct effect of IVE on fire blanket is reported, however later it is exposed that no significant effect of IVE on behavior was found. Please clarify this.

I think here the problem may arise because the total effect is non-significant while the direct effect is significant when the mediator is partialized. By checking the estimates, it is possible to see, as the authors indicated, that indirect and direct effects have opposite signs. This, together with the fact that the direct effect is bigger than the total effect is an indication of an inconsistent mediation, where the mediator acts as a suppressor variable in the relationship between IVE and blanket. Given that the mediator is part of the model, this may be interpreted as a significant direct effect when the mediator is taken into account, and should be discussed accordingy.

Pp 36. A nonsignificant effect is presented as "marginally significant". I suggest avoiding using this language on the manuscript because they can be misleading.

Pp 37. Please provide assumption checks for the repeated measures anova, and report results in a corresponding table.

Discussion:

The authors propose here that the relationship between IVE and self-efficacy is partly mediated by vulnerability, however in their model (figure 3) they estimate a co-variation and not an indirect effect properly. Please clarify this point, and in the case there is an indirect effect, please report it accordingly. If no mediation effect was computed, this should be not discussed as such.

Please provide a substantial interpretation of the significant mediation effect of IVE to blanket towards knowledge, so the reader can get an idea of the mechanism involved.

The authors discuss the nonsignificant interaction effect. In this case, it is advisable to avoid discussing nonsignificant results as “marginally significant”. The authors describe this as a “fully” mediated effect, which may not be the case given what I exposed before.

The authors discuss the importance of causal claims; however, this is beyond the scope of the present article (I describe this at the beginning). Same for the influence of IVE on the psychological domains, it is important to control for previous values.

General comments:

The authors usually justify their decisions with statements such as "We feel it is appropriate or at least worthwhile" (pp 7), "we felt it was relevant to compare the IVE experience with an information sheet". Although their decisions seem reasonable from a methodological point of view, they need to be justified based on methodological or theoretical grounds (i.e. benefit of simultaneous measurement of self-report and behavior, benefit of a control condition versus no control condition).

Please state the difference between serious and commercial games.

I hope the authors find this review useful for their research!

Best regards,

6. PLOS authors have the option to publish the peer review history of their article (what does this mean?). If published, this will include your full peer review and any attached files.

Reviewer #1: No

Reviewer #2: Yes: Cristóbal Hernández C.

---

## [Author Response · Author response to Decision Letter 0]

30 Oct 2019

PONE-D-19-17627

Playing with fire. Understanding how experiencing a fire in an immersive virtual environment affects prevention behavior.

PLOS ONE

Dear Ms. Jansen,

Thank you for submitting your manuscript to PLOS ONE. After careful consideration, we feel that it has merit but does not fully meet PLOS ONE’s publication criteria as it currently stands. Therefore, we invite you to submit a revised version of the manuscript that addresses the points raised during the review process.

We would appreciate receiving your revised manuscript by Oct 03 2019 11:59PM. To enhance the reproducibility of your results, we recommend that if applicable you deposit your laboratory protocols in protocols.io, where a protocol can be assigned its own identifier (DOI) such that it can be cited independently in the future. For instructions see: http://journals.plos.org/plosone/s/submission-guidelines#loc-laboratory-protocols

• A rebuttal letter that responds to each point raised by the academic editor and reviewer(s). This letter should be uploaded as separate file and labeled 'Response to Reviewers'.

• A marked-up copy of your manuscript that highlights changes made to the original version. This file should be uploaded as separate file and labeled 'Revised Manuscript with Track Changes'.

• An unmarked version of your revised paper without tracked changes. This file should be uploaded as separate file and labeled 'Manuscript'.

We look forward to receiving your revised manuscript.

Kind regards,

Geilson Lima Santana, M.D., Ph.D.

Academic Editor

PLOS ONE

Journal Requirements:

For this resubmission we converted the figure files via PACE. 

For the rest, we thought that we adhered to all PLOS ONE’s style requirements. If we missed something, can you please point us to the specific style criteria that we have to adjust?

OK

In the interest of full disclosure, we wish to draw your attention to the following: the first author, Patty Jansen, works three days per week as a marketing researcher at an insurance company (Achmea) and two days per week at Eindhoven University of Technology on her Ph.D. research. Her employer contributes partly to her Ph.D. research by financially supporting her for one day per week. The employer also financially supported this research. We declare that the research that we present was in no way influenced by the insurance company. The funder had no role in study design, data collection and analysis, decision to publish, or preparation of the manuscript.

https://www.achmea.nl/

We note that you received funding from a commercial source: Achmea

We included a Competing Interests Statements. 

4. "We note from your Cover Letter and Financial Disclosure that the first author of the manuscript (FCPJ) is affiliated to an insurance company. Please update the affiliations within your Cover Page and within Editorial Manager to reflect this affiliation.

We included the affiliation within the Cover Page. 

We do however not understand how the information at “current address” differs from the “affiliation” information if postal codes or street addresses are not allowed. Also, we were not totally sure if we correctly used the provided symbols. If we made any mistake here, please let us know so that we can correct it. 

Reviewers' comments:

Reviewer's Responses to Questions

Comments to the Author

1. Is the manuscript technically sound, and do the data support the conclusions?

Reviewer #1: Partly

Reviewer #2: Partly

2. Has the statistical analysis been performed appropriately and rigorously? 

Reviewer #1: Yes

Reviewer #2: No

3. Have the authors made all data underlying the findings in their manuscript fully available?

Reviewer #1: No

Reviewer #2: Yes

4. Is the manuscript presented in an intelligible fashion and written in standard English?

Reviewer #1: Yes

Reviewer #2: Yes

5. Review Comments to the Author

We thank both reviewers for the constructive comments, and have chosen to implement almost all of them. We copied and numbered the reviewers’ questions and wrote our answers directly below the questions of the reviewer. The page numbers correspond with the file ‘Manuscript’. 

Reviewer #1: I’ve read the paper with interest. It’s about a timely topic (comparing VR and non-VR approaches for teaching safety behaviors) and aims at exploring an additional variable with respect to other studies.

The way the paper is written does not yet meet the requirements of an high-quality presentation, so I provide a number of comments in the following about necessary improvements, that I hope we will be helpful in preparing a thoroughly revised version.

First, the paper needs to tone down its initial claims of extreme originality (three of the four claims are exaggerated).

“Our study has four characteristics that, taken together, make it stand out from most previous IVE studies.” COMMENT: Actually this claim is not accurate and must be toned down for three of the four characteristics.

1. “First, our study focuses on the fire domain, while most risk related IVE-effect studies

involve other risk domains (e.g. health, traffic safety, environmental risk, aircraft evacuation).” COMMENT: Several IVEs concerning fires have actually been built, see this scholar search as a starting point:

https://scholar.google.com/scholar?hl=en&as_sdt=0%2C5&q=%22virtual+reality%22+Fire&btnG=

“Second, we consider the effects of IVE on a full set of “psychological determinants

(knowledge, vulnerability, severity, self-efficacy, and locus of control). “ COMMENT: Also effects of virtual risk experiences on such variables have been previously studied (and the paper should contrast its research with the other findings on these psychological variables), for example:

Chittaro L., Designing Serious Games for Safety Education: "Learn to Brace" vs. Traditional Pictorials for Aircraft Passengers, IEEE Transactions on Visualization and Computer Graphics, 22, 2016, 1527-1539.

Chittaro L., Sioni R., Serious Games for Emergency Preparedness: Evaluation of an Interactive vs. a Non-Interactive Simulation of a Terror Attack, Computers in Human Behavior, 50, 2015, 508–519.

Both these papers also refer to the same theoretical basis of the submitted manuscript (Protection Motivation Theory, PMT)

“Third, we compare the effects of IVE with a control condition which allows establishing the effect of IVE over and above a more standard way of getting the same information across.” COMMENT: See above

“Fourth, and most importantly, we measure actual prevention behavior, and test to what extent psychological determinants have an effect on actual behavior.” COMMENT: This is the actual key point to stress. It’s the characteristic on which this research stands out.

Response to Reviewer comment No. 1:

Just for argument’s sake, we would like to mention that our statement “Our study has four characteristics that, taken together, make it stand out from most previous IVE studies.”, is actually correct if taken to mean that there aren’t that many studies that take all four characteristics into account. We agree with the reviewer though, in the sense that it can easily be read as meaning to say that each of them separately is new. 

We toned down the text and changed our text as follows. First, we have removed the argument of the control group as indeed several more IVE-studies have used a non-interactive control. We think the other two arguments still hold, but need more explanation, which we have now added (see p. 3). 

[Application to fire prevention]: Several IVE’s concerning fires have been built, but they typically are used to study human behavior in the case of a fire or they serve as training purposes for fire men or to teach fire drill skills to children. What we meant to say is that this is the first IVE-effect study in the fire domain, in which the IVE serves as a means (through the simulation of fire event) to induce changes in the psychological determinants, and subsequently change behavior. We have now added this argument to the text (p. 3). 

[Full set of psychological determinants]: Previous studies have indeed studied these variables before, and we also refer to studies of Chittaro in our paper. We were however not aware of the papers that the reviewer mentioned, and thank the reviewer for the suggestions. We have now integrated the given references in our paper (see ref 15 and 18). What none of these studies do, however, is analyze the effects of the IVE on these variables simultaneously. This is one beneficial feature of our Structural Equation Modelling approach. We now elaborate on this in the text (p. 3). 

2. However, the authors have to be careful about terminology concerning behavior. The two behaviors measured in the study were 1) if a participant would invest part of his or her show-up fee in a fire blanket and 2) whether a participant would take home flyers related to fire safety. There’s a terminological issue all over the paper (that can however be easily fixed): the paper uses the term “prevention behaviors” to refer to the two behaviors, but what they actually show is a possible interest in prevention (for example, taking home a flyer is no guarantee that the participant will keep and read the flyer and take the preventive actions described in the flyer).

Response to Reviewer comment No. 2:

We partly agree with the reviewer here. Investment of the show-up fee in the fire blanket is an observed behavior as well as taking home flyers. We refer to it as behavior in contrast to the more standard form of self-reported behavior (e.g. “I have the intention to buy a fire blanket”). We believe the term “possible interest”, as suggested by the reviewer, does not really cover the purchase of a fire blanket, which is a real prevention measure. But we agree that a flyer in itself does not reduce the probability nor severity of a fire. This was actually already noted in our Method section: “The primary measurement for the effect of IVE on prevention behavior will be the participants’ purchase of a fire blanket, and the secondary measurement is the participants’ interest in fire prevention information.” (p. 25). We have added a remark about the taking home of the flyer being a rather ‘soft’ kind of behavior in the text on page 25. 

3. The paper (correctly) stresses the importance of VR presence creating the expectation that the construct will be measured in the study, but then it is not. This is a bit confusing.

Response to Reviewer comment No. 3:

We do not think the concept of presence was relevant in this study for comparing the conditions, as we compared an IVE with an INFO condition. The INFO condition was not expected to have any effect on “presence” and questions about the level of presence concerning the information sheet would have been confusing. For evaluation purposes we did however include one item after the IVE about the level of realism of the IVE (see p. 19, p. 27). 

4. According to PMT, a simple hypothesis that Perceived Vulnerability and Perceived Severity positively influence behavior, as the paper does (H3b and H4b) is not straightforwardly supported. The theory indeed clearly states that if vulnerability and severity are perceived, but the recommended behavior is not presented in a way that convinces the message recipient about efficacy (both recommendation efficacy and self-efficacy), then the influence will be negative instead of positive: the recipient will engage in emotion control behaviors instead of the recommended risk control (prevention) behavior.

Response to Reviewer comment No. 4:

We agree with the reviewer that the behavior -owning a fire blanket- must be perceived as effective or useful in order to be performed. The IVE stresses the response efficacy of the fire blanket, as the viewer observes the fire being extinguished by the use of the fire blanket, but we did not measure the perceived response efficacy. Although this could have been a useful addition (as mentioned in the limitation section on p. 44), we did not want to prime participants in the direction of a fire blanket by asking specific questions about the fire blanket. We added this explanation on p. 44. 

5. Hypothesis “H5c: Perceived self-efficacy negatively influences prevention behavior” needs more theoretical motivation. As it is, it apparently contradicts the theory (PMT) on which the paper aims at being substantially grounded.

Response to Reviewer comment No. 5:

Self-efficacy in our paper refers to the confidence people have in their own capacity to correctly handle the situation of a grease fire without having a fire blanket (only participants without a fire blanket were recruited). People often downplay the severity of a fire, overestimate the time they have to handle the fire, and overestimate their own capabilities (e.g. remain calm and use the lid of a pan). This might result in taking less prevention measures. 

This is comparable to the road safety domain, in which a higher level of self-efficacy (over estimation of one’s driving capabilities) is related to more unsafe behaviors and accidents. It is common in this domain to develop interventions to decrease someone’s perceived self-efficacy in order to enhance safe behaviors. We integrated several additional references in the paper that show the relationship between self-efficacy and unsafe driving behaviors (p. 15). 

An alternative that we rejected, but perhaps would have been closer to other applications of PMT, would have been to try to increase self-efficacy as defined by someone’s confidence to be able to extinguish the grease fire with a fire blanket. This would have been more appropriate when we would have been aiming at explaining the actual use of a fire blanket, instead of buying it. In addition, in the current IVE the participant just had to focus their eyes a couple of seconds to the fire blanket to “extinguish the fire with the fire blanket”. To really increase self-efficacy related to using the fire blanket in the IVE, special gloves should ideally be used to realistically simulate gestures of picking up the blanket and putting it on the grease fire. To show the participant how it should be done, and increase one’s self-confidence in using the fire blanket successfully. 

Because we wanted to focus on the purchase of the fire blanket, we chose to try and decrease the level of self-efficacy (overestimation of correctly handle a grease fire situation), as is common in the road safety domain, in order to stimulate the need for extra prevention measures (as mentioned on p. 15, 16). 

6. The description of the IVE condition is lacking of enough details. For example, the paper says “The participant can then choose between different actions: 1.) go towards the source of the fire (the kitchen) and extinguish the fire with the fire blanket. The fire will be extinguished and the game ends. Or 2.) go towards…”

A first question is how such actions are selected? Second, the granularity of the actions is not clear. For example, what happens if one selects action 1? All those listed things happen as the user passively watches? Or does the user have to perform them? In this latter case, there are multiple actions. And what does the user do in the physical world to perform the multiple actions?

More generally, since one of the goals of the IVE in the paper is to increase users’ knowledge, it is important to fully describe everything users could do and what was the feedback from the environment, also highlighting the pedagogical aspects of the IVE design.

Attaching videos to the submission could provide some additional help.

Response to Reviewer comment No. 6:

We agree with the reviewer that we could have been clearer about the IVE and the possible scenarios. We have now included more details in our text and added a detailed description of each scenario in the S2 file (p. 20-22). 

The participant can move through the environment (with a joystick) and can perform different actions by focusing on objects with his eyes (e.g. open doors, pick up a toddler, use fire blanket). So the participant does not passively watch, but does everything him / herself, albeit with relative simple controls. 

On YouTube one can see a video of a similar IVE, a version made that was made for the Samsung Gear (in which a smartphone has to be placed), although there are some differences with the one used in the experiment. In the Samsung Gear version people do not use a controller to move through the IVE, but instead look at arrows to move in a certain direction. Also, the environment is less immersive than the one of the ORDK. But it might nevertheless be useful to get an impression: https://www.youtube.com/watch?v=digXSxrrjV0

7. In the hardware section, the paper says the software was implemented on one

desktop machine and one laptop. But what happened during the study? Some people used the laptop and some other the desktop?

Response to Reviewer comment No. 7:

Yes, we had one laptop in one room, and the desktop in the other room. Participants were randomly assigned to one of the rooms. We changed the text so that it is clear that we used one laptop and one desktop device during the study (p.22).

8. The measurement section is not always clear too. The first measurement is said to “include current prevention behaviors that are beyond the score (sic) of this paper.” What does this mean? Were people asked about their fire prevention behaviors?

Then the paper says “The prevention behaviors that were measured in the first and third measurement are not analyzed in this paper”. One can try to guess that this was done to assess if there was a behavior change (at least a self-reported ones). If this is the case, it’s strange that the paper omits to analyze it, since the main difference from other research in the literature concerns specifically participant’s behavior.

Response to Reviewer comment No. 8:

In an earlier paper we have studied whether prevention behaviors from various domains (burglary, water damage, fire) can be considered to form a one-dimensional scale (using Rasch analysis). The general conclusion was that these behaviors indeed form a one-dimensional scale. We included all these previously measured prevention behavior items in the first and third measurement, to be able to replicate our previous findings and to see whether the person ordering that the Rasch scale provides, remains consistent under the VR intervention. In this sense, these extra measurements consider the issue of the appropriateness of the Rasch scale measurement, much less the effect of VR. Given that the aim and topic of this measurement was different and our paper already lengthy, we did not include it.

9. To measure the SLOC construct, the paper creates a questionnaire in which some items are supposed to measure internal SLOC while the others external SLOC. However, the section does not report a factor analysis to confirm that the set of created items can actually be divided in the two subscales.

Response to Reviewer comment No. 9:

Based on literature we expected 2 factors. In our SEM we found that the two underlying factors had high correlation, which suggests that we cannot really divide SLOC in two underlying subscales. To confirm this finding we ran a EFA that does indeed shows the ISLOC and ESLOC items load on to different factors, but with many items showing cross loadings and high correlations (>.5 between the two factors). We can supply the results of this analysis, if necessary. NB Other papers differ in whether they treat SLOC as one or two scales: in this sense, finding that both sub-dimensions cannot be distinguished is not rare.

10. One of the items (“How much panic do you think there will be in case of a grease fire?”) that were used to measure severity is different from the usual items that measure that construct, and the choice should be explained.

Response to Reviewer comment No. 10:

De Hoog, Stroebe and de Wit (2008) measured severity with three items, namely how severe, harmful, and serious respondents perceived the health consequences of hypoglycemia. We copied the harmful and severity items. However, the term “serious” translated into Dutch is however also severe (in Dutch “ernst”) or can be translated as “taking something seriously” (in Dutch “serieus”). The latter explanation is however not very relevant for a grease fire. With a health issue such as hypoglycemia someone can take the symptoms and consequences seriously or not. However, it is unlikely to not take a grease fire seriously: anyone will agree that some kind of action needs to be taken. We changed the item to panic this this also reflects the severeness of a fire situation. We now added this explanation in the paper (p. 24).

11. One of the items (“People should be rewarded by their insurance company if they take

preventive measures to prevent or control fire”) that were used to measure SLOC does not actually concern locus of control orientation.

Response to Reviewer comment No. 11:

This item was based on two items of the Aviation Safety Locus of Control Scale (Hunter, 2002), namely:

• Pilots should lose their license if they periodically neglect to use safety devices (for example, seat belts, checklists, etc.) that are required by regulation. (ISLOC)

• Pilots should be fined if they have an accident or incident while “horsing around”. (ISLOC)

The items of the Aviation Safety Locus of Control Scale are negatively formulated: lose license or be fined if they do not follow safety procedures, and our item is positively formulated: people get rewarded if they take safety precautions, but otherwise this is a direct translation of this item. 

12. Unlike the first two questionnaires, the third questionnaire was administered on-line. The paper does not provide details about actions taken to prevent issues of on-line surveys, for example: were the answering times measured and taken care of ?

Response to Reviewer comment No. 12:

Whereas in our case we could expect answers to be somewhat more reliable than in an unsolicited survey (people knew beforehand the study also included a third on-line), we tried to take into account the potential unreliability of surveys as much as possible. The survey tool we used (www.mwm2.nl) measures the survey completion time. This allows to exclude those with very small completion times. For longer times matters are less straightforward. When someone does not close the browser, the time will keep on running, so the average or “upper” completion times need not be an indication of unreliability. In the Table below we included the registered completion times per survey. Although the second and the third survey do not exactly include the same items, they are comparable in length (42 resp. 47 items). The minimum completion time in the second survey was 220 seconds and in the third survey 223 seconds, and also the percentage in the lowest time category is similar. This gave us no reason to believe that participants hastily completed the third online survey. 

 First survey Second survey Third survey

0-299 60.3%* 9.9%** 8.8%***

300-599 39.7% 74.8% 51.9%

600-899 0 14.9% 20.8%

>900 0 .4% 18.5%

N 242 242 216

* The minimum completion time was 157 seconds

** The minimum completion time was 220 seconds

***The minimum completion time was 223 seconds

Besides this, we checked the data for inconsistencies and for instance for multivariate outliers with the BACON algorithm using Stata 14 and did not find any (p. 26). 

13. The “descriptive statistics” says that the effect of the IVE is smaller than expected, but provides no effect sizes in the statistics.

Response to Reviewer comment No. 13:

We do not really understand this point and feel we do treat effect size in the text. We indicated that the effect is smaller than we assumed when calculating our sample size: we expected a 20 percentage point difference (assuming 20% for the INFO group vs 40% for the IVE group) (as was mentioned on p. 18). In the experiment we found a difference of 8.6 percentage points (39.8% for the INFO group vs 48.4% for the IVE group) (p.27). We now include the expected difference in the descriptive statistics paragraph (p.27). 

In the case the reviewer refers to a lack of mention of effect size measures such as Cohen’s: these are usually provided when testing mean differences between continuous variables. In this case, Cohen’s d can be used in order to overcome the fact that continuous variable distributions can differ in both location and scale, so that the same difference in means could be either large or small, depending on the variation. But probabilities do not have this issue: the difference between the probabilities in the groups is the effect size. We can nevertheless calculate Cohen’s d here and it equals -.172. We did not change the main text, but can add this as desired by the reviewer.

14. In Table 1, the translations from Dutch to English is not always good, in one case “Fire in a home usually exists by the people themselves” the reader cannot be sure of what that is supposed to mean.

Response to Reviewer comment No. 14:

We agree with the reviewer and rewrote some of the items (p.30, 31). 

15. The percentage of participants who felt nauseous during the VR experience is surprisingly high and needs discussion. It’s an indication that some VR design guidelines were likely violated by the implemented IVE. This can limit significantly the effectiveness of the IVE for the goal intended in the paper.

Response to Reviewer comment No. 15:

Cyber sickness is a common phenomenon with IVE’s and similar to regular motion sickness, and many individuals suffer from this (cf. Weech, Kenny & Barnett-Cowan, 2019; McCauley & Sharkey, 1992 although both do not give exact estimates). Causal factors can be related to the design of the IVE such as the visual display characteristics (e.g. frame rate), but also to IVE’s in general such as the sensory mismatch between the observed world and the virtual world, or related to the persons such as gender and gameplay experience. 

The specifications of the Oculus Rift DK2 Head-Mounted Display (HMD) were:

Resolution: 960 x 1080 per eye 

Field of view: 100 degrees diagonal 

Latency: 40 ms (estimation of the developer)

Display frame rate: 60 FPS (estimation of the developer)

These specifications are consistent with other VR research and do not point to a violation in the VR design. 

As mentioned on p.33 we have tested all the relationships with SEM also on the sample minus the participants who stated they became nauseous during the IVE experience (n = 215). Results showed that all estimated relationships remained stable, indicating that the level of nausea did not affect the results. To give an indication: when performing a chi-square test on the dataset without the nauseous participants, results are similar: in the INFO group 39.8% chose the fire blanket compared to 48.5% (χ2 (1, N = 215) = 1.609, p = .205).

To conclude, although the percentage of nauseous people perhaps might appear high, cybersickness is a very common phenomena (especially when any kind of discomfort can be evaluated as such) and the specifications of the IVE give no reason to doubt violation of the guidelines. Furthermore, the analyses give no argumentation for potential different effects when nausea is avoided.

Weech, S., Kenny, S. & Barnett-Cowan, M., (2019). Presence and Cybersickness in Virtual Reality Are Negatively Related: A Review. Frontier Psychology, 10: 158.

McCauley, M.E. & Sharkey, T.J. (1992). Cybersickness: Perception Virtual Environments. Presence, 1(3). 

16. The result that participants using VR learned less than those using printed information needs much more discussion. From the details and screenshots provided, a simple and likely explanation can be traced back to the design and implementation of the IVE, which seems too primitive compared with the state of the art in the VR literature. The paper should discuss the differences in interactivity, realism, and pedagogical methods between the IVE used and the more sophisticated IVEs available in the literature.

Moreover, since the IVE can influence psychological determinants both indirectly (through level of knowledge) and directly, the limitations in the IVE can have had consequences that reach furthest than knowledge only, and weaken the conclusions of the paper. This should be discussed in a critical way.

Care must also be taken in the conclusions section which currently seems to assume that the results hold for IVEs in general, while the limitations in the studied IVE make it difficult to support generalization to more complex IVEs used in learning and training.

Response to Reviewer comment No. 16:

We agree with the reviewer that there are more technically advanced IVE’s, also ones in which people can really move (HTV Vive) or that trigger multiple senses (e.g. heat, wind, fragrance). The IVE was developed in 2014/ 2015 and the study was conducted in 2015, and since then more advanced IVE’s have obviously been available. At the time of the experiment, most of our participants never experienced an IVE (with a head mounted display) and were actually quite impressed by the experience. The mean score for the level of realism of the experience was 3.82 and 3.43 (on a 5 point scale) for the severity of the virtual fire. 

The literature that we are referring to used IVE’s of a similar level of interactivity and realism (to our opinion): Chittaro (2012; 2014; 2015; 2016), Zaalberg and Midden (2013), Ahn, Bailenson and Park (2004). Given that previous studies, with IVE’s that are equally or even less advanced (e.g. Chittaro, 2010), have delivered positive results before, and given that most of our subjects had not been introduced to VR before, we had no reason to assume the necessity of a much more elaborate IVE. 

However, we do agree with the reviewer that the way in which the knowledge was integrated in the IVE was not optimal, as we had mentioned on p. 40: “An unexpected result was that people in the INFO condition scored higher on knowledge. This could possibly be explained by the fact that the information in the IVE was presented as text in the head-mounted display, which may not have been so easy to read and was perhaps also not compatible enough with the rest of the virtual experience.” 

And in line with this remark, on p.40: “One could imagine that the effect of the IVE intervention could be improved by changing the way in which information is put forward. For instance, improved hardware and software might make it easier to produce a static text that is easier to read than the one in our IVE was.”

We integrated a suggestion of how to better integrate knowledge in an IVE as done by Chittaro (2015; 2018). We integrated this on p. 40.

Of course, it is certainly possible that a more technically advanced IVE that reflects a fire even more realistically by better graphics, heat, real smoke etc produces more anxiety and will have larger effects. 

If we understand the reviewer correctly, he or she suggests that knowledge might influence vulnerability and severity etc as well, which might affect the conclusions of the analysis. This is, however, not a relation that we have seen in the literature before, nor had we hypothesized it before (although one might be able to come up with an argument along those lines), so we do not feel comfortable adding arrows to the theoretical model as we have it. In addition, mod indices in our SEM models did not suggest such arrows. We can and now do nevertheless address this issue in the discussion (p. 41).

Reviewer #2: In the present manuscript, the authors provide an interesting approach towards fire prevention by using an immersive virtual environment procedure to teach people how to react in the case of a grease fire. The inclusion of actual prevention behavior, together with self-report measures and the use of an experimental setting with randomization and control condition gives this article a promising potential. However, there are some writing and data-analytic issues that, in my opinion, need to be addressed before a recommendation for publication. In my review, I will focus mostly in consistency, writing form and methods which is my area of expertise.

1. The statistical analyses (SEM) done in this article do not take full advantage of the experimental setting created by the authors, given that the results are not controlled by baseline levels (as in a group x time interaction model), which is a problem the authors should address in detail. It is a common practice to measure the constructs in a pre-post fashion, however the authors did not do it to avoid priming. However, the trade-off is that there is no baseline control anymore, turning the analyses correlational, which should be addressed by the authors in more depth.

Response to Reviewer comment No. 1:

We agree that the trade-off between having a baseline and making sure there is no priming is of key importance and in our submission we have argued explicitly about it (p. 44). However, the reviewer’s point that this “turn[s] the analyses correlational” we do not see. Indeed, the analyses are correlational, but they would have been correlational too if we had included a pre-intervention measurement. Both with and without a pre-intervention measurement, the study remains experimental in the sense that the intervention was randomly allocated across participants. Moreover, the extra measurement comes at the expense of additional measurement error, so even the gain in precision that a within-subject manipulation might give, is not guaranteed (see http://datacolada.org/39). We’re not sure whether we should address this point in more detail in the text than we have, but we have reworded the text to make this point clearer and can further elaborate on these issues if wanted, unless the reviewer is suggesting something else here that we apparently miss.

2. On the other hand, I find the data analysis and results section rather confusing. This is, in my opinion, mostly because it is very data dense and contains a lot of modelling, which always creates a challenge in reporting. One consequence of this, is the omission of important information to evaluate the quality of the data analyses. This is especially relevant for the CFA and SEM parts, where they show some inconsistencies that I’m listing below.

pp 6 (and others). When talking about psychological determinants, please be more specific. Psychological determinants of prevention behavior?

Response to Reviewer comment No. 2:

We included a sentence on p. 3 to make this clear for the remainder of the paper. 

3. pp 7: "The underlying arguments are equally appropriate for the area of fire prevention behavior" It is important to give arguments when presenting such a statement, why?

Response to Reviewer comment No. 3:

We now include this argumentation on p. 7. 

4. Pp 7. "Studies usually use" is not the best argument for selection of variables: I strongly suggest the authors to describe a sound logical and empirical basis for their proposal. This is done later; however, it can be briefly explicated here.

Response to Reviewer comment No. 4:

We cannot find a sentence with “studies usually use” in our manuscript. We think the reviewer referred to this sentence:

“IVE studies often also consider knowledge about the topic and locus of control as variables that may be influenced by IVE and may themselves influence subsequent prevention behavior”. 

In any case, we agree with the reviewer and have adjusted the text (p. 7). 

5. Pp.7 "We feel it is appropriate or at least worthwhile to study psychological determinants and the target behaviors simultaneously" Why? It is, indeed, important, but this should be elaborated.

Response to Reviewer comment No. 5:

We now elaborate on this point (p. 8).

6. pp 8. "Can in fact be studied empirically relatively easily" How?

Response to Reviewer comment No. 6:

We were referring to studies in the aircraft evacuation domain (e.g. Chittaro 2012; 2014; 2015; 2016) in which it is not that easy (to say the least) to measure whether people follow the correct safety procedures during an accident. In the case of fire prevention measures, measuring the real behavior that we consider is way easier, since we can just measure the purchase of the fire blanket. On the other hand, measuring if people will follow the correct safety procedures during a fire, indeed faces the same problems as in the aircraft evacuation domain. We decided the remove the sentence (p. 8).

7. pp11. Please check repeated citation (29,29).

Response to Reviewer comment No. 7:

Now corrected (p. 11). 

8. pp11. Please provide references for the SLOC bi-dimentionality, if they were provided before, I suggest moving the definition of the construct together with the citations above.

Response to Reviewer comment No. 8:

Now integrated (p. 10). 

9. pp 22. Why are only knowledge, vulnerability and severity5 included in the third measurement? This exclusion should be supported.

Response to Reviewer comment No. 9:

We now explain this exclusion (p. 23). 

10. Measures: Please report basic psychometric properties of the instruments here (cronbach's alpha for internal consistency, for example, and factor structure of original instruments).

Response to Reviewer comment No. 10:

The psychometric properties (Cronbach’s alpha, factor loadings and AVE) for the eventually chosen items/ scales are given in Table 1 on p. 30. We also supplied the data for the process of removing items that do not fit well, but based on AVE (following the procedure as in Knijnenburg &Willemsen, 2015), not based on factor loadings or Cronbach’s alpha. We added an Appendix to this document (“Appendix for reviewers”) in which the factor loadings and Cronbach’s Alpha’s are presented for the original scales (Table 1). We can add this Table to the paper if considered useful according to the reviewer. 

Knijnenburg, B.P., Willemsen, M.C. (2015). Evaluating recommender systems with user experiments. In: Ricci, F., Rokach, L., Shapira, B. (eds.) Recommender Systems Handbook, pp. 309–352. Springer, New York. 

11. pp. 27. Please specify the software version used for SEM analyses.

Response to Reviewer comment No. 11:

Now included (p. 28). 

12. Data analyses and report:

Please describe which sampling wave was used for the analyses (I guess it’s sample 2, however I cannot find it in the manuscript).

Response to Reviewer comment No. 12:

It is indeed sample 2. This was mentioned on p. 23:

“The hypothesized model (Fig 1) is based on differences between the IVE and the INFO group at the time of the second measurement.” 

13. Measurement models: I find it problematic to test all measurement models together, given that this would further obscure local sources of bad fit. Is it just one instrument the one creating the bad fit? Are all measurement models wrong? I would suggest test this separately. On the other hand, the procedure of co-variation between items within an instrument should be done in this step.

Response to Reviewer comment No. 13:

Opinions differ on whether estimating the model in its entirety or estimating the parts separately is the better approach. We followed the procedure to do CFA on multiple factors in one single estimation as suggested Brown (2006) and performed by Knijnenburg and Willemsen (2015). 

Brown, T. A. (2016). Methodology in the social sciences. Confirmatory factor analysis for applied research. New York, NY, US: The Guilford Press.

Knijnenburg, B.P., Willemsen, M.C. (2015). Evaluating recommender systems with user experiments. In: Ricci, F., Rokach, L., Shapira, B. (eds.) Recommender Systems Handbook, pp. 309–352. Springer, New York. 

Nevertheless, we have re-run our models with separate measurement models for each factor individually, and for the accompanying modified models. See the enclosed Appendix, Table 2-5 for the model fit statistics and Table 6-7 for the R-squared estimates and AVE’s. It should be noted that when performing CFA on individual factors with only 3 indicators this shows a perfect fit because these are “just-identified” (df=0) (see Table 2-3). When performing CFA’s on the individual factors similar results apply compared to testing the measurement models all together, and the same items should be removed to improve AVE. The Appendix is quite lengthy and offers little extra value to the paper, but we can of course include it if wanted. 

14. Structural Equation Models: Modifications were made to the models, which is a step towards a good fit. It is important to see if the modifications suggested by the modification indexes are reasonable under theoretical grounds. This is done for the co-variation between variables; however, it should also be made for the covariation between items. It is usually accepted to co-vary items within a scale if they share method or another known source of shared variance that should not be captured by the latent factor (please see previous comment, however, regarding the location of this procedure).

Response to Reviewer comment No. 14:

We did not have any theoretical grounds for adding the covariation between items: correlation between any of the items in principle is not unreasonable though. The main reason to add them was to create maximum fit model and see whether this impacts on our results. As mentioned on p. 32: “We only incorporated this step to show how model fit could be improved, but will not elaborate on this step, as adding correlations at the item level is not very common.” We indeed found that adding these correlations does not affect our conclusions. We now left them in, but we can also leave model 3 and 5 out of the paper, as desired the reviewer. 

15. It is not clear why do the authors base their report on model 4, if the model with a good fit is model 6. It is important to report the good fitting model. Even though models showed no substantial changes in their point estimates, the non-significant results are part of a bad-fitting model and thus are not directly interpretable (i.e. SLOC). This should be at least stated as a limitation or tested with a different regression technique.

Response to Reviewer comment No. 15:

The models that we present represent different trade-offs between on the one hand theoretical rigor (what did we hypothesize beforehand) and empirical fit (which covariations can or should we take into account to make sure the measurement model fits as best it can). We chose Model 4 for our conclusions as the one that remains closest to our a priori theoretical arguments and nevertheless has a decent fit but, as can be seen in the paper, reporting the results based on Model 6, as the reviewer suggests, is possible too and causes no changes in the statistically significant effects (and of course has better fit statistics). 

We do not see why the non-significant results are the necessary consequence of a poorly fitting model (as opposed to showing that there is no true effect), given that Models 4 and 6 lead to essentially the same results, although we could report Model 6 in the text instead (we did not do this now). Perhaps we are misunderstanding the reviewer here, or perhaps the reviewer mixed up some things. The SLOC variable that the reviewer mentions is not in the estimated model at all. 

16. I find it important to explicitly say, on the SEM models, how were the dichotomous outcomes treated, and interpret them accordingly. Where they estimated using a probit or logistic regression? (In Mplus this is different for different estimation methods)? Please clarify this on the manuscript.

Response to Reviewer comment No. 16:

We used Weighted Least Squares, which goes with probit regression in M-Plus (maximum likelihood would have used logistic regression). We added this information on p. 28. 

17. Please also describe which method did the authors use to compute the mediation analyses (multiplicative?) and provide the estimates in the tables.

Response to Reviewer comment No. 17:

Indirect effects in Mplus are indeed calculated by multiplying the direct effects. The results of this, including the bootstrapped results, are in running text (p. 37). Do you want us to provide the results in the text and in a separate table?

18. Pp 36. Please specify which bootstrap method was used. Bias corrected, percentile? They have shown different trade-offs regarding type 1 errors.

Response to Reviewer comment No. 18:

We used the bootstrap percentile method. We now mention this in the text on p 37.

19. Pp 36. A direct effect of IVE on fire blanket is reported, however later it is exposed that no significant effect of IVE on behavior was found. Please clarify this.

I think here the problem may arise because the total effect is non-significant while the direct effect is significant when the mediator is partialized. By checking the estimates, it is possible to see, as the authors indicated, that indirect and direct effects have opposite signs. This, together with the fact that the direct effect is bigger than the total effect is an indication of an inconsistent mediation, where the mediator acts as a suppressor variable in the relationship between IVE and blanket. Given that the mediator is part of the model, this may be interpreted as a significant direct effect when the mediator is taken into account, and should be discussed accordingy.

Response to Reviewer comment No. 19:

The interpretation by the reviewer is indeed correct. Perhaps the confusion arose because one could argue that there is a positive direct effect when controlling for the mediator. Instead we describe our findings in terms of the total effect (none) and two separate significant effects: a positive direct effect and a negative indirect effect through the mediator. We now clarified that the total effect we found is non-significant at p. 37.

20. Pp 36. A nonsignificant effect is presented as "marginally significant". I suggest avoiding using this language on the manuscript because they can be misleading.

Response to Reviewer comment No. 20:

Now adjusted on p. 37.

21. Pp 37. Please provide assumption checks for the repeated measures anova, and report results in a corresponding table.

Response to Reviewer comment No. 21:

Good point. We see now that the assumption for normally distributed residuals was violated (checked with Shapiro-Wilk test). We therefore performed the analyses again, using a multilevel regression analysis instead, with a robust estimator correcting for the non-normally distributed residuals. Results show some changes in significances, as reported on p. 38, 39. The conclusions about the interaction effects remain the same. 

We provide the changed results in the text. We did not really get which results the reviewer wanted us to report in a table (what is left to report?), but are willing to do so, if necessary.

22. Discussion:

The authors propose here that the relationship between IVE and self-efficacy is partly mediated by vulnerability, however in their model (figure 3) they estimate a co-variation and not an indirect effect properly. Please clarify this point, and in the case there is an indirect effect, please report it accordingly. If no mediation effect was computed, this should be not discussed as such.

Response to Reviewer comment No. 22:

We think this might be a misunderstanding. We propose the relationship between IVE and self-efficacy is partly mediated by severity, not vulnerability, and we estimated it as such. In figure 3 we have single headed arrows for relationships between concepts and double headed arrows for correlations between concepts. These correlations between concepts arise from the mod-indices. 

23. Please provide a substantial interpretation of the significant mediation effect of IVE to blanket towards knowledge, so the reader can get an idea of the mechanism involved.

Response to Reviewer comment No. 23:

We integrated an interpretation of the mediation effect of knowledge (as a mediator between IVE - fire blanket), and we added an explanation for the missing mediation effect (knowledge as a mediator between IVE- flyers, as was hypothesized) (p.41, 42). 

24. The authors discuss the nonsignificant interaction effect. In this case, it is advisable to avoid discussing nonsignificant results as “marginally significant”. The authors describe this as a “fully” mediated effect, which may not be the case given what I exposed before.

Response to Reviewer comment No. 24:

We agree. We maintained the paragraph that discusses the direct and indirect results, but changed the wording in both this paragraph and the discussion section in a way that reflects the suggestion by the reviewer (p. 37, 41, 42).

25. The authors discuss the importance of causal claims; however, this is beyond the scope of the present article (I describe this at the beginning). Same for the influence of IVE on the psychological domains, it is important to control for previous values.

Response to Reviewer comment No. 25:

Please see our comment in the beginning, where this point comes up before: having an earlier measurement (“previous values”) would have been possible but we deliberately did not do this as we outlined above. Nevertheless, the design remains experimental: participants were randomly divided over 2 conditions.

General comments:

26. The authors usually justify their decisions with statements such as "We feel it is appropriate or at least worthwhile" (pp 7), "we felt it was relevant to compare the IVE experience with an information sheet". Although their decisions seem reasonable from a methodological point of view, they need to be justified based on methodological or theoretical grounds (i.e. benefit of simultaneous measurement of self-report and behavior, benefit of a control condition versus no control condition).

Response to Reviewer comment No. 26:

We added justifications in the text (p. 8, 9). 

27. Please state the difference between serious and commercial games.

Response to Reviewer comment No. 27:

We think the reviewer is referring to this sentence:

“Kato, Cole, Bradlyn, and Pollock (2008) have shown a positive effect of a 3-month serious game on knowledge about cancer treatment, compared to a control group who received a commercial game.”

In this case the commercial game was the video game Indiana Jones and the Emperor’s Tomb. 

We integrated this, and also provided a definition of a serious game (p. 9).

I hope the authors find this review useful for their research!

We thank the reviewers for their time, effort, and useful comments.

Best regards,

6. PLOS authors have the option to publish the peer review history of their article (what does this mean?). If published, this will include your full peer review and any attached files.

Do you want your identity to be public for this peer review? For information about this choice, including consent withdrawal, please see our Privacy Policy.

Reviewer #1: No

Reviewer #2: Yes: Cristóbal Hernández C.

---

## [Editor Report · Decision Letter 1]

23 Dec 2019

PONE-D-19-17627R1

Playing with fire. Understanding how experiencing a fire in an immersive virtual environment affects prevention behavior.

PLOS ONE

Dear Ms. Jansen,

Thank you for submitting your manuscript to PLOS ONE. After careful consideration, we feel that it has merit but does not fully meet PLOS ONE’s publication criteria as it currently stands. Therefore, we invite you to submit a revised version of the manuscript that addresses the points raised during the review process.

We would appreciate receiving your revised manuscript by Feb 06 2020 11:59PM. To enhance the reproducibility of your results, we recommend that if applicable you deposit your laboratory protocols in protocols.io, where a protocol can be assigned its own identifier (DOI) such that it can be cited independently in the future. For instructions see: http://journals.plos.org/plosone/s/submission-guidelines#loc-laboratory-protocols

We look forward to receiving your revised manuscript.

Kind regards,

Geilson Lima Santana, M.D., Ph.D.

Academic Editor

PLOS ONE

Additional Editor Comments (if provided):

Thank you for incorporating reviewers's advices.

I believe it is important to include the appendix to reviewers. This would help interested readers have a deeper understanding of the methods and results found.

Please, in your manuscript, you need to include not only page numbers, but also line numbers. Use continuous line numbers (do not restart the numbering on each page).

---

## [Author Response · Author response to Decision Letter 1]

27 Jan 2020

1.) We added the appendix to the manuscript, labeled as S4 File (as agreed to by e-mail by Sarah Mills, on Jan 24th 2020).

2.) We added line numbers in the manuscript.

---

## [Editor Report · Decision Letter 2]

3 Feb 2020

Playing with fire. Understanding how experiencing a fire in an immersive virtual environment affects prevention behavior.

PONE-D-19-17627R2

Dear Dr. Jansen,

We are pleased to inform you that your manuscript has been judged scientifically suitable for publication and will be formally accepted for publication once it complies with all outstanding technical requirements.

With kind regards,

Geilson Lima Santana, M.D., Ph.D.

Academic Editor

PLOS ONE

---

## [Editor Report · Acceptance letter]

21 Feb 2020

PONE-D-19-17627R2 

Playing with fire. Understanding how experiencing a fire in an immersive virtual environment affects prevention behavior. 

Dear Dr. Jansen:

I am pleased to inform you that your manuscript has been deemed suitable for publication in PLOS ONE. Congratulations! Your manuscript is now with our production department. 

With kind regards,

on behalf of

Dr. Geilson Lima Santana 

Academic Editor

PLOS ONE